



# An adaptive regional vulnerability assessment model: Review and concepts for data-scarce regions

Mark Bawa Malgwi[1,2], Sven Fuchs[3], Margreth Keiler[1,2,4]

1. University of Bern, Institute of Geography, Hallerstrasse 12, 3012 Bern, Switzerland

2. University of Bern, Oeschger Centre for Climate Change Research, Hochschulstrasse 6, 3012 Bern, Switzerland

3. University of Natural Resources and Life Sciences, Institute of Mountain Risk Engineering, Peter-Jordan-Str. 82, 1190 Vienna, Austria

4. University of Bern, Mobiliar Lab for Natural Risks, Hallerstrasse 12, 3012 Bern, Switzerland

*Correspondence to*: Mark Bawa Malgwi (mark.malgwi@giub.unibe.ch)

**Abstract**

Although the vulnerability indicator method has been applied to several data-scarce regions, a missing linkage with damage grades had hindered its application for loss evaluation to complement disaster risk reduction efforts. To address this gap, we present a review of physical vulnerability indicators and flood damage models to gain insights on best practice. Thereafter, we present a conceptual framework for linking the vulnerability indicators and damage grades using three phases (i)
developing a vulnerability index, (ii) identifying regional damage grades, and (iii) linking vulnerability index classes with damage grades. The vulnerability index comprehensively integrates elements of the hazard using a Building Impact Index (BII) on one hand, and exposure, susceptibility and local protection elements using a Building Resistance Index (BRI) on the other hand. For the damage grades, local expert assessments are used for identifying and categorizing frequently observed regional damage patterns. Finally, by means of synthetic what-if analysis, experts are asked to estimate damage grades for
each interval of the BII and class of BRI to develop a vulnerability curve. The proposed conceptual framework can be used for damage prediction in data-scarce regions to support loss assessment and to provide guidance for disaster risk reduction.

**Keywords**: Data-scarce regions, vulnerability indicator, damage grade, floods, building, disaster risk reduction

## 1  Introduction

The magnitude and frequency of floods and their impact on elements at risk has increased globally (Quevauviller, 2014).
Risks associated with floods are especially high for communities with limited capacity to resist impacts. Communities with low resistance to impacts of hazards are often referred to as vulnerable. Although the definition of vulnerability varies in



different fields of study, efforts aimed at understanding and reducing vulnerability are regarded as important steps for disaster risk reduction (UNISDR, 2015). UNISDR (2009) defined vulnerability as the conditions that make communities susceptible to the impact of hazards. These conditions may be linked to limited access to resources, to missing risk transfer mechanisms, and to poor housing quality if elements at risk are considered. Focusing on the latter, poor housing quality has been shown to be a key factor if different regions exposed to the same hazard level are compared (Papathoma et al., 2003; Keiler et al., 2006). Although the vulnerability of a community has social, economic, physical, environmental, institutional and cultural dimensions (Birkmann et al., 2013), these dimensions are connected (Mazzorana et al., 2014). Fuchs (2009) and Papathoma-Köhle et al. (2011) identified physical vulnerability as a prior state that impacts other dimensions of vulnerability. WHO (2009) highlighted that there is a strong connection between physical flood vulnerability with social and economic vulnerability dimensions, pointing out that the disruption of physical elements directly affects social and economic activities within a society. Additionally, Blanco-Vogt and Schanze (2014) reported that the impacts of floods on the built environment may trigger migration of people from areas at high risk to low-risk areas. As a result, an increasing number of studies focus on the understanding of physical vulnerability since it directly influences social and economic dimensions. Physical vulnerability assessment supports evaluation of economic losses (Blanco-Vogt and Schanze, 2014), analysis of physical resilience (Papathoma-Köhle et al., 2011), cost-benefit evaluation (Holub and Fuchs, 2008), risk assessment for future scenarios (Mazzorana et al., 2012), and decision-making by stakeholders responsible for hazard protection through e.g. resource allocation (Fuchs, 2009).

Common approaches used for assessing physical vulnerability to flood hazards include vulnerability functions or curves, matrices, indicators (Papathoma-Köhle et al., 2017) and more recently multivariate methods. Vulnerability curves, often also called stage-damage curves, show the relationship between flood depth and the degree of impact (e.g., damage grades, relative or absolute loss). These curves are developed using empirical data or expert knowledge. The empirical method requires data on flood depths and related building damage or losses after a flood event. These data allow to search for suitable curves to correlate flood depths to damage or losses. Synthetic methods utilize a what-if analysis based on expert knowledge to determine expected damage for selected intervals of flood depths. Generally, vulnerability curves use flood depths as the only damage-influencing variable. This has been identified as a major limitation for these curves since flood damage is also influenced by several other parameters relating to both, the hazard and building characteristics (Thieken et al., 2005; Schwarz and Maiwald, 2007). Vulnerability matrices are based on qualitative descriptions to relate a hazard magnitude to an impact pattern. Multivariate methods deduce relationships between empirical building damage or loss data and multiple damage-influencing parameters by means of statistical techniques. Thus, including several damage-influencing parameters relating to building and hazard characteristics is a major advantage of the multivariate method compared to vulnerability curves. However, due to underlying data requirements, difficulties exist in applying the multivariate method in data-scarce regions. Such data-scarcity has been identified to hinder risk assessment which consequently restricts disaster risk reduction



efforts (Niang et al., 2015; Englhardt et al., 2019). Vulnerability indicators are based on aggregated variables to
communicate the state of a system (e.g. the resistance of a building) and to provide insights in the level to which this system
will be impacted by a certain hazard level (Birkmann, 2006). A vulnerability index is obtained using deductive (e.g. expert
knowledge), inductive (e.g. data) or normative (e.g., interview with community residents) approach. Since vulnerability
indicators do not require empirical data, it has gained increasing popularity and usability in data-scarce regions.

The need for further research to identify vulnerability indicators for hazard impact assessment has been emphasized by the
Hyogo Framework for Action 2005-2015 (ISDR, 2005). More recently, several studies have re-emphasized the importance
of identifying and understanding vulnerability indicators as a fundamental step in disaster risk reduction (e.g., UN/ISDR,
2015; Zimmermann and Keiler, 2015; Klein et al., 2019). Vulnerability indicators allow the identification of parameters that
influence how hazards impact the built environment. These impacts are usually measured in terms of damage or loss. The
focus on using vulnerability indicators is particularly encouraged in data-scarce regions because of a low demand for
empirical data (Papathoma-Köhle et al., 2017). In addition, indicators are more-easily communicable to decision makers
(Birkmann, 2007) and they have application in several stages of the risk cycle (Papathoma-Köhle et al., 2017). The
vulnerability indicator method has been applied for assessing different natural hazards and risks, including studies on
earthquakes by Rashed and Weeks (2003), Yücemen et al. (2004), Schmidtlein et al. (2011), Brink and Davidson (2015),
Peng (2015), and Marleen et al. (2017), and on landslides by Silva and Pereira (2014), Guillard-Gonçalves et al. (2016), and
Thennavan et al. (2016). Applications for tsunamis include studies by Papathoma-Köhle and Dominey-Howes (2003),
Dominey-Howes and Papathoma-Köhle (2007) and Dall'Osso et al. (2009), on debris flows by Kappes et al. (2012),
Rheinberger et al. (2013), Thouret et al. (2014), Papathoma-Köhle et al. (2017), and studies on flood hazards by Balica et al.
(2009), Akukwe and Ogbodo (2015), and Carlier et al. (2018). However, these studies differ with respect to vulnerability
definitions, vulnerability dimensions, the spatial scale of application, the applicability in the risk cycle, and according to the
function of the developed index. Detailed reviews on vulnerability assessment methods are given by Papathoma-Köhle et al.
(2017) and Fuchs et al. (2019a)..

Vulnerability assessment methods are mainly used to estimate damage or loss. Kircher et al. (2006) highlighted that damage
to buildings is the most important factor for estimating economic and social loss. Consequently, an increased amount of
studies have investigated the impacts of hazards and how to predict building damage (Ke et al., 2012). Such predictions of
building damage are carried out using flood damage models. Flood damage models (or loss models) show the relationship
between the extent of building damage on one hand and hazard and building characteristics (damage influencing factors) on
the other hand. Generally, flood damage models are developed using empirical data or synthetic (what-if) approaches (Merz
et al., 2004, Naumann et al., 2009; Merz et al., 2010; Maiwald and Schwarz, 2015; Chen et al., 2016). A further
classification of flood damage models can be differentiated based on models that predict damage using a single damage-



influencing parameter (e.g., Penning-Rowsell et al., 2006; Naumann et al., 2009) and models that use multiple damage-influencing parameters (e.g., Thieken et al., 2008; Maiwald and Schwarz, 2015). Flood damage models can predict damage patterns or grades(e.g., Maiwald and Schwarz, 2015; Ettinger et al., 2016) or monetary value of such damage (e.g., Thieken et al., 2008; Merz et al., 2013). Since damage grades represent qualitative descriptions of frequently observed damage patterns with a region (moisture defects, cracks on supporting walls), they provide a good basis for damage estimation and

enhance comparability of flood impacts between flood events, regions, and between buildings types (Blong, 2003a). In addition, since damage grades are comparable for similar building types (Maiwald and Schwarz (2015), they improve transferability of flood damage models (Wagenaar et al., 2017). Reviews on flood damage models can be found in Merz et al. (2010), Jongman et al. (2012), Hammond and Chen (2015), Romali et al. (2015) and Gerl et al. (2016).

Developing flood damage models require empirical data either during model development or for checking model
performance. However, due to scarcity of empirical data in many data-scarce regions, limitations occur in developing these models. Consequently, data scarcity has been identified to hindered disaster risk reduction efforts in such regions (Niang et al. 2015). More recently, Englhardt et al. (2019) reemphasised data-scarcity as the limiting factor in vulnerability assessment in developing countries. Moreover, owing to difference in regional building types, limitations exist in the transfer of damage prediction models from data-rich to data-scare regions (Tarbotton et al., 2015; Englhardt et al., 2019). For example, studies
by Cammerer et al. (2013) and Maiwald and Schwarz (2015) have demonstrated that empirically-deduced damage models only allow a transfer of the damage distribution and vulnerability models in study areas with close agreements to hazard and building characteristics. Thus, there is a strong need for developing further approaches for evaluating the vulnerability of buildings to flood risk in data-scarce regions.

Only little has been known so far on the vulnerability and damage mechanism of buildings exposed in developing countries,
such as in Africa. Adelekan et al. (2015) identified populations and assets in African cities to be among the most vulnerable globally. Consequently, with climate change, the number of extreme events and catastrophic impacts in vulnerable regions are expected to increase (Mirza, 2003). In Africa particularly, the need to develop a systematic approach in evaluating preconditions of buildings and how they are impacted by flood hazards has been stressed by stakeholders and researchers (Komolafe et al., 2015). Although, sandcrete block and clay buildings are the most predominant building types in many
African countries (Gasparini, 2013), flood damage models remained underdeveloped for such building types (Komolafe et al., 2015). Commonly, exposure and vulnerability are mainly assessed in a regional context based on very coarse data and aggregated land-use classes resulting in very high uncertainties of vulnerability, especially in the rural context (de Moel et al., 2015). A recent study by Englhardt et al. (2019) presented a new approach including an object-based analysis of buildings and their properties (construction type, building material) based on exposure data from GFDRR (2018) to deduce
vulnerability curves. The results provide insights in the influence of different datasets for exposure and hazard models.





However, the information of exposure is aggregated in grid cells (15" x 15"), and building types and the related stage-damage curves were extracted from available literature, which in combination with the low spatial resolution resulted in high uncertainties. Thus, beside such recent studies addressing flood exposure und vulnerability in data-scarce areas, there is a strong need for developing further approaches for vulnerability and risk assessments in these regions.

Papathoma-Köhle et al. (2017) recommended a combination of vulnerability assessment methods to take advantage of their individual strengths while minimizing their weakness. Such combination of methods, particularly expert-based approaches, might provide a desirable compromise for data-scarce regions. For example, Godfrey et al. (2015), using Romania as a case-study, reported a combination of methods to develop an approach for data-scarce regions. However, applicability of the method has shown to be restricted to regions where vulnerability curves for typical building types already exist. In addition,
because of a limited sample size used to test the method results may be biased (Godfrey et al., 2015).

Approaches using indicators and damage grades are in general more suitable for data-scarce areas, yet, so far there is a gap in systematically linking them. The aim of this paper is to develop a conceptual framework for assessing flood vulnerability of the build environment in data-scarce areas, such as in Africa. To do this, we first provide a review on vulnerability indicators for flood, as well as an overview of flood damage models. We combine the gained knowledge from the review to develop a
conceptual framework that links vulnerability indicators and flood damage grades by utilizing local expert knowledge. A special focus for the conceptual framework is on transferability and adaptation to specific regional conditions.

This paper is structured as follows: Section 2 provides an overview of literature with a critical evaluation of individual elements of vulnerability indicators, followed by a review of methodological steps of constructing vulnerability indicators. In Section 3, a brief review of flood damage models is presented. While Section 4 addresses the need for linking indicators and
damage grades, Section 5 introduces the conceptual framework for the linkage as well as the steps for operationalizing the framework. Discussions and conclusions are presented in the final section.

## 2    Review of indicators for building vulnerability assessment

The use of vulnerability indicators to assess regional vulnerability is becoming increasingly popular (Papathoma-Köhle et al., 2017). This increase is likely due to the link indicators provide between research, practice and political need (Hinkel,
2011). Hence, indicators serve as a good communication tool to stakeholders or communities exposed to hazards. Another reason for increased popularity applying indicators is a low requirement of empirical data. In addition, vulnerability indicators supplement the use of vulnerability curves and matrices in a way that the overall picture on flood vulnerability becomes clearer. This clarity is achieved by an integration of multiple drivers of vulnerability providing a more holistic perspective of vulnerability-contributing factors. Fuchs et al. (2011) noted that holistic evaluation of vulnerability requires



the inclusion of all its dimensions (e.g. social, physical, environmental, economic). However, a model that will incorporate all these dimensions will be quite complex. As a result, evaluations of vulnerability are usually carried out for specific dimensions of vulnerability. In this study, we focus on indicators developed for assessing physical vulnerability to floods with a specific attention to buildings.

## 2.1 Overview of indicators for flood hazards

Prior to developing indicators, the framework has to be set. This framework includes a variety of elements (we refer to these as indicator elements) which helps to clearly outline the extent of applicability of the derived vulnerability index. The framework also helps to transparently communicate the spatial extent within which the index is still valid. Basic elements defining the framework of a vulnerability index include the study aim, the region of application, the vulnerability dimension, the approach, the application in risk cycle, the spatial scale and the index output. For each indicator element, we present an

overview from literature (Table 1) for different types of floods and evaluate its specific function. Although further studies exist regarding flood vulnerability indicators, the studies summarized in Table 1 were selected due to their relevance in addressing building vulnerability. Common elements important for framing indicator schemes will be analysed in the following sections.

### 2.1.1 Indicator aims

A first step in developing a framework for indicators is to define the aim; this means clearly defining what the indicators (or derived index) are to be used for. Table 1 outlines the different aims of constructing vulnerability indicators for different flood types. Generally, the aim of indicators is to simplify a concept through the use of indices (Heink and Kowarik, 2010; Hinkel, 2011). Indicators aim to give facts about the present state of a system and support decision-making in risk reduction by aggregating information on an element at risk (Günther, 2006). Using components representing building, human and

economic vulnerability, Papathoma et al. (2003) carried out an investigation aimed at assessing the vulnerability of coastal areas to tsunami hazards using an indicator approach, later referred to as the PTVA (Papathoma Vulnerability Assessment Model). However, in a subsequent revision of the PTVA, a modification in the overall aim of the index to assess building vulnerability to tsunami impact resulted in the exclusion of the human and economic dimensions, allowing an in-depth evaluation of parameters that directly influence building damage to tsunami (Dall'Osso et al. 2009). Furthermore, the

application of the index was extended to allow the assessment of probable maximum loss. Indicated by this example, we have to consider by developing vulnerability indicators to set first the aim of the vulnerability assessment so that methods and data can be better tailored to the overall aim of the study. This is the reason why indicators that serve to identify hotspot areas either for emergency planning or mitigation purpose draw inference from multiple vulnerability dimensions like social





and physical (e.g. Kienberger et al., 2009; Akukwe and Ogbodo, 2015), while others developed for damage evaluation focus
on the physical dimension (e.g. Dall'Osso et al., 2009; Godfrey et al., 2015).

### 2.1.2 Region of application

Indicators are adaptive to a regional context, hence, a set of indicators selected for a particular region is not necessarily
transferable to another region (Papathoma-Köhle et al., 2017, 2019). Barroca et al. (2006) emphasized that indicators are
context-dependent and it will be ineffective to adopt an indicator, or a set of indicators, that are non-existent in a target
region. For example, several vulnerability indices use the number of storeys in assessing the impact of floods to buildings
(e.g., Kappes et al., 2012; Blanco-Vogt and Schanze, 2014; Thouret et al., 2014; Godfrey et al., 2015; Fernandez et al., 2016;
Sadeghi-Pouya et al., 2017). The underlying theory is well-founded based on the prevalence of multi-storey buildings in
these regions and from an engineering standpoint since the additional weight of more storeys helps in resisting the impact
from flood. Consequently, it is important to highlight the region where the indicator is applicable. In addition, providing
information with regards to the characteristics of the built environment (e.g. building typology) as well as hazard types can
also serve as a basis with which other regions can assess the functionality of the indicators in other regions.

### 2.1.3 Vulnerability dimension

The interactions between different vulnerability dimensions (physical, social, economic, environmental) generate challenges
for assessing vulnerability. Although some studies focus on one specific dimension of vulnerability when assessing
indicators (e.g. Dall'Osso et al., 2009; Blanco-Vogt and Schanze, 2014; Bagdanavičiute et al., 2015; Godfrey et al., 2015),
other studies focus on multiple dimensions of vulnerability (e.g. Kienberger et al., 2009; Balica, et al., 2012; Krellenberg and
Welz, 2017). Table 1 shows that the majority of vulnerability indicators in the context of flood are multidimensional.
Birkmann (2006) noted that that the choice of carrying out a multidimensional study might be related to data availability
since in some countries, these data are readily available. However, given the complexity of the vulnerability concept,
difficulties exist regarding how comprehensive multidimensional studies can be due to challenges arising from having too
many indicators (Cutter and Finch, 2008). In addition, care must also be given on how to systematically aggregate different
vulnerability dimensions.

### 2.1.4 Application in the risk cycle

An asset of vulnerability indicators is that they may be used in different stages of the risk management cycle, multiple
applications include disaster preparedness, response and mitigation (Papathoma-Köhle et al., 2017). Vulnerability indicators



help to identify important parameters influencing the impact of flood on the built environment, hence, emergency planners can identify highly vulnerable areas and prioritize efforts during disaster preparedness and response. For mitigation purpose, decision-makers or community residents can improve the resistance capacity of identified indicators (e.g. building material, construction quality) to reduce hazard impacts. Indicators facilitate the implementation of risk reduction strategies by identifying vulnerability parameters. As shown in Table 1, however, most applications of vulnerability indicators are for preparedness and mitigation purpose. In other studies, indicators are used to estimate building damage potential, hence providing a basis for monetary loss estimation (e.g. Dall'Osso et al., 2009; Blanco-Vogt and Schanze, 2014).

### 2.1.5 Spatial scale

Spatial scale is identified to be an important issue in evaluating impacts of hazards (Birkmann, 2007; Fuchs et al., 2013; Kundzewicz et al., 2019). The spatial scale for applying the vulnerability indicator approach varies depending on the availability of data (Marleen et al., 2017) or the aim of the assessment. Spatial scales for assessing vulnerability can be on micro-, meso- and macro-scale, however, Table 1 shows that vulnerability indices for assessing physical vulnerability to flood hazards is mostly applied at a local (micro) scale. Local scale, in this case, refers to either individual building scale (e.g. Papathoma et al., 2003; Kappes et al., 2012;  Blanco-Vogt and Schanze, 2014), or to semi-aggregated scales such as entire building blocks (e.g. Müller et al., 2011; Krellenberg and Welz, 2017; Sadeghi-Pouya et al., 2017) or neighborhoods (e.g. Fernandez et al., 2016; Percival et al., 2018). Local-scale assessment is usually challenging in terms of data collection (Günther, 2006), in particular in less-developed countries with missing metadata on land-use, exposure and population. However, other indicators operate on a smaller meso or macro scale, and they are mostly implemented using Geographical Information Systems (GIS). Small-scale assessments can serve to give an overview of vulnerability (hotspot assessment) on a larger area, hence, decision makers are able to use them in allocating resources for emergency response or risk mitigation. Messner and Meyer (2006) stated that while meso-scale assessments deal with regional boundaries, macro-scale assessments focus on national or international boundaries, each approach calling for different interpretation (Eriksen and Kelly, 2007).

### 2.1.6 Index output

Index output refers to the aggregated state of the indicators. Since vulnerability is complex and contextual, some studies choose to frame vulnerability not in absolute values, but relatively (e.g. Dall'Osso et al., 2009; Kappes et al., 2012; Sadeghi-Pouya et al., 2017). In this way, the vulnerability index serves as a mean to compare between elements of the built environment, expressing which one is more or less vulnerable. However, outside the context of making comparisons, this output might lack a significant independent meaning (Tarbotton et al., 2015). While some studies formulate a vulnerability index by directly aggregating identified indicators, others develop sub-indices as a preliminary to allow for separate



assessment of individual components that contribute to vulnerability. For instance, using a set of multiple indicators, Balica et al. (2009) developed sub-indices for physical, social, environmental and economic vulnerability. In the next step, they aggregated these sub-indices to provide a multidimensional composite index for assessing flood vulnerability. An advantage of the sub-indices is that it supports policymakers who are only interested in a specific component of vulnerability. Nonetheless, due to the complex interactions of different vulnerability components, Hinkel (2011) recommended that their
aggregation should not be purely subjective but supported by data from previous hazard events.

## 2.2 Constructing a vulnerability index

Constructing a vulnerability index involves selecting, weighting and aggregating of indicators. The index carries information on the extent to which an element can be impacted by a hazard, given the combined influence of selected indicators. Figure 1 shows steps and corresponding outputs and methods commonly applied for constructing a vulnerability index for flood
hazards. Different methods used in deriving the index include deductive (based on theories/basic assumptions), inductive (based on empirical data) and normative (based on value judgment) approaches. In physical vulnerability assessment to flood hazards, the deductive approach is the most commonly applied method relying on expert judgment and information provided in the relevant scientific literature without any further empirical data. It is also common to use a combination of inductive and deductive approaches either during the indicator selection or during indicator weighting and aggregation. Nevertheless,
prior to selecting, weighting and aggregating indicators, a framework should be developed to address how major components of the indicator fit together (Birkmann, 2006; JRC, 2008). The framework should clearly communicate basic data requirements, hazard type applicability, vulnerability dimension, and index generation. In this section, we look at each step of deriving the vulnerability index. Table 2 shows different studies that derived a vulnerability index to assess flood hazards and various methods employed.

### 2.2.1 Indicator selection

The core element in deriving a vulnerability index is the indicator itself (Krellenberg and Welz, 2017), hence, careful attention has to be given to which variables are chosen as indicators. The selection of indicators is one of the main challenges of vulnerability assessment (Marleen et al., 2017; Papathoma-Köhle et al., 2019) because a suboptimal selection of indicators will consequently lead to an information bias or even loss (Günther 2006). Before a variable is qualified to be an indicator,
certain criteria have to be considered to allow for consistency and methodical soundness. Important criteria for selecting a variable to be an indicator are issues related to measurability, relevance, analytical and statistical soundness (see Birkmann 2006; JRC, 2008 for a complete list of criteria for indicator selection). Selected indicators should provide good guidance in order to capture how an element will be impacted (e.g. building damage) by a phenomenon (e.g. flood). Since capturing





vulnerability is a complex task, multiple indicators are usually required for an objective evaluation. However, since the aim
of indicators is to reduce complexity, attention should be given to achieving a balance between the number of indicators
selected and the reduction of complexity (Günther, 2006; Barroca et al., 2008). From a practical point of view, it is usually
easier to manage a sizable number of indicators which also results in less time needed for data collection and associated cost
implications.

The selection of vulnerability indicators can be categorized into two phases; a preliminary and a final selection phase (cf.
Table 2). In the preliminary selection phase, an initial selection of a range of identified variables is carried out. This serves to
identify all possible variables from e.g. literature or expert knowledge that influence vulnerability. As shown in Fig. 1, the
preliminary selection is carried out either using a deductive or normative approach. In the final phase, the number of
variables to be used for weighting or aggregation is reduced. The final selection can be based on data availability, statistical
analysis, expert opinion or other evaluation procedures. For example, Kienberger et al. (2009) reported a spatial vulnerability
assessment tool using the indicator approach. In their study, expert knowledge was used for the preliminary selection of
indicators. Thereafter, based on structured rounds of questionnaire evaluation, a final selection was made based on a Delphi
approach. The Delphi approach utilizes several indicator suggestions by different experts and combines the suggestions after
a consensus is reached through several rounds of questionnaire exchange. During the Delphi process, pre-selected indicators
that are identified to be less relevant are removed in order to arrive at a set of more effective indicators. The Delphi approach
can be applied for selecting both primary indicators and their classes (e.g. building material as an indicator having classes of
masonry, wood and reinforced concrete). In another study, Müller et al. (2011) used a combination of literature review,
expert opinion, and suggestions by household owners in the study region for preselecting vulnerability indicators. However,
the final selection of indicators was based on expert weighting through establishing a cut-off weight to determine which
indicators are to be removed or selected. Another approach recently becoming popular in physical vulnerability assessment
is based on Principal Component Analysis (PCA). After pre-selecting indicators, a final selection is achieved based on PCA
to reduce the dimensions of a data set to a number of dimensions required to describe the variance of the data set (JRC,
2008). Examples of studies which were based on PCA for final indicator selection include Akukwe and Ogbodo (2015) and
Fernandez et al. (2016), see Table 2. In some of these studies, however, the final indicator selection and weighting of
indicators overlap.





**2.2.2  Indicator weighting**

After the selection of indicators, the next step is to assign weights to allocate the extent to which each indicator is relevant with respect to the targeted vulnerability assessment. Birkmann (2006) pointed out that the assignment of weights is what makes an indicator out of a variable. Prior to assigning weights to different indicators, a scoring is assigned for components of an indicator, for example, building type as an indicator can have reinforced concrete, masonry and wooden buildings as

sub-components: we refer to these sub-components as indicator classes. The scoring of these indicator classes is further based on internal weighting since in each case the score for each indicator class is based on individual vulnerabilities observed. For example, it is common to assign reinforced concrete building a score that assumes a lower vulnerability to flood impact compared to masonry or wooden buildings if we assume a similar hazard magnitude (e.g., building vulnerability classification by Maiwald and Schwarz, 2012).

The weighting of indicators and scoring of indicator classes can be carried out using deductive, inductive, or normative approaches.

1. The deductive approach is based on research-based knowledge and conclusions of previous studies. The weighting is based on deduction, or inference from frameworks, set of concepts, or theories on vulnerability (Hinkel, 2011). Different types of deductive weights basically include direct expert weights, expert weights in combination with

literature analysis or the application of an Analytical Hierarchy Process (AHP) from expert knowledge.

Firstly, direct expert weights refer to weights assigned to indicators using the knowledge of experts either by questionnaires or interviews. Approaches based on direct expert weights are among the most common types of weighting in physical vulnerability assessment; examples for floods include studies by Kappes et al. (2012), Sadeghi-Pouya et al. (2017) and Carlier et al. (2018), see Table 2. The often observed subjectivity of experts has,

however, initiated some critical debates on this method. Questions such as "Who qualifies as an expert?", "How many experts are needed for an objective assessment?" have been raised. As a result, Hinkel (2011) referred to expert judgment as a rather weak form of deductive argument which should only be used for the selection of indicators.

Secondly, some vulnerability studies used weights from literature in combination with expert knowledge to

formulate new weights to indicators. For instance, Blanco-Vogt and Schanze (2014) used weights from previous studies as a basis for subsequent expert weighting to assess the physical vulnerability of building material to floods. In another study, Krellenberg and Welz (2017) investigated the probability of buildings to be exposed under certain socio-environmental conditions using the indicator approach by combining literature review and expert opinion for assigning weights to selected variables. Weighted variables from this study included building structure, building

surrounding and coping capacity. However, in order to use weights from a literature review as a basis for weighting indicators, there should be comparable regional settings and hazard typologies.

Thirdly, a commonly applied weighting method for physical vulnerability assessment is based on the AHP, a multi-criteria decision tool which has clear advantages over the direct weighting techniques (Saaty, 1980). The AHP allows the breakdown of a complex problem into smaller components through a pair-wise comparison system.

Instead of evaluating each indicator relative to all other indicators based on an objective weighting consideration, using AHP the weighting is carried out between pairs of indicators. The pair-wise comparison allows an evaluation of which indicator in every pair is more important than the other using a scale of 1 (equal importance) to 9 (extreme



importance) (Chen et al., 2012). The decision on which indicator is more important can be evaluated from analysing data or expert knowledge, however, the expression of the extent to which indicator is more important than another is basically based on expert knowledge. For example, in a pairwise comparison for normally built reinforced concrete and clay buildings, we can use damage data to assign a weight indicating that reinforced concrete buildings have lower vulnerability compared to clay buildings, even if the decision to assign a vulnerability level may be quite subjective. To ensure minimal subjectivity in a pairwise comparison, the Consistency Ratio (CR) can be computed. The CR checks if the subjectivity of comparisons are within an allowable limit, and once the condition of CR is not fulfilled, a repetition of the process has to be carried out (Golz, 2016). Studies that applied the AHP include those of Kienberger et al. (2009) and Godfrey et al. (2015), see Table 2. For example, Godfrey et al. (2015) used the AHP for weighting indicators and scoring indicator classes, and showed the consistency ratio in each case. In addition to having an application with both quantitative and qualitative data, another advantage of the AHP is the ability to track errors through the consistency ratio. Depending on the total number of indicators, however, the AHP can be computationally demanding. JRC (2008) pointed out that weights from the AHP should be interpreted more as trade-offs between indicators and not as a direct measure of importance, a conclusion that was also drawn by Mazzorana and Fuchs (2010). Dodgson et al. (2009) highlighted a lack of internal consistency and lack of theoretical basis in the 1-to-9 scoring system as a limitation of the AHP which is still an ongoing discussion between fellows (DETR, 2009).

2.  Another weighting approach is the inductive approach, using indicator weights from conclusions based on analysing observations. Inductive weights utilize inference from data (Hinkel, 2011). In physical vulnerability assessment, the PCA is the main method employed for extracting inductive weights. The PCA technique uses linear combinations to explain the variance in a data set (JRC, 2008) by reducing the dimensions of the data set to few components that account for most of the variance in the data. The PCA initializes a procedure whereby weights (factor loadings) are assigned to the indicators based on their variance in the original data set. Studies that use the PCA includes Thouret et al. (2014), Akukwe and Ogbodo (2015) and Fernandez et al. (2016), see Table 2. JRC (2008) and Hinkel (2011) argued that PCA results should be interpreted carefully since the underlying theory of the approach does not directly reveal the relationship between the variables and the phenomena. Chow et al. (2019) further emphasized that without a proper standardization of the data sets, resulting factor loadings from PCA would be misleading since the approach does not directly take into account the response variable. Consequently, the use of a direct and measurable proxy for vulnerability, for instance, building damage grade or loss ratio, is important to standardize variables before carrying out the PCA. JRC (2008) pointed out that the PCA is sensitive to the size of variables, number of observations, the presence of outliers and to changes in the original dataset.

3.  Another form of weighting which is not very common in physical vulnerability assessment is the normative approach. Using the normative approach weights can be assigned based on value judgment (Hinkel, 2011). The normative approach is based on priorities of individuals. A common application of the normative approach is the equal weighting approach; based on a value judgment, all variables influencing vulnerability are taken to be equally important (Frazier et al., 2014). However, JRC (2008) argued that equal weighting results in the same level of importance for individual indicators which raises the question which indicators are more important with respect to vulnerability. Adopting an equal weights approach is sometimes required in cases where no consensus is reached on a suitable weighting alternative. Few studies that applied the equal weights approach include those of Balica et al. (2009), Behanzin et al. (2015) and Ntajal et al. (2016), see Table 2. In studies where multiple dimensions of vulnerability are considered, the equal weights approach will favour dimensions with a higher number of indicators



if an unequal number of indicators is used. However, such irregularities can be corrected by a systematic
normalization. Furthermore, Chen et al. (2012) noted that the equal weighting approach cannot properly handle
indicators that are highly correlated because these are double-counted. Another implication of the approach,
particularly at the aggregation phase, was noted by Hinkel (2011): Equal weighting means all indicators are ideal
replacements of one another, and low values in one indicator can be compensated by high values in another
indicator. Another example of the use of value judgment for weighting indicators was demonstrated in Müller et al.
(2011), whereby household owners were asked to weight selected indicators. As shown in Table 2, this method is
not very popular in physical vulnerability assessment.

### 2.2.3   Indicator aggregation

Physical vulnerability assessment incorporates different types of indicators with different units of measurement, for example,
building material and distance to the hazard source. Therefore, before aggregating different indicators, it is necessary to find
a systematic and consistent means of representing the indicators while retaining their conceptual meaning. A first step
towards achieving a fair representation of different indicator types for aggregation is to scale the indicators. Qualitative or
quantitative indicators are mostly adapted to an ordinal scale whereby data are categorized using an increasing or decreasing
order, this order should be based on previous research results. The difference between the categories in the ordinal scale is
usually non-uniform, but mostly subjective to fit the indicator framework. A good example of the use of the ordinal scale
was demonstrated by Dall'Osso et al. (2009) where five categories were used to scale all selected indicators. Other studies,
however, do not provide such details of data transformation techniques used. Asadzadeh et al. (2017) noted that scaling of
indicators is sensitive to normalization and aggregation output, hence, it is important to adopt a scaling that fits the data and
the overall vulnerability framework

Aggregation of indicators refers to a systematic combination of indicators to create an index. Generally, several methods for
indicator aggregation exist, however, a commonly applied method for physical vulnerability assessment is the additive
method (see Table 2). This method is based on a summation of the product of the weights and scores (or the scaled value) of
all selected indicators. The summation can be directly on scores of the indicators (direct additive method) or after applying
weights to the scores (weighted additive method).  The result of the indicator aggregation is influenced by the applied
aggregation technique as some approaches allow to counterbalance indicators with low values (compensation).  In the
additive method, a constant level of compensation for lower values is allowed. For example, the high indicator value of a
building with poor construction material can be compensated with a low indicator value because the building is located at a
far distance from the river channel. If an equal weighing is applied in combination with a direct additive aggregation method,
it will mean all indicators are perfect substitutes (Chen et al., 2012). Other methods for aggregation include the geometric
and multi-criteria method (JRC, 2008), however, these methods are not usually applied in physical vulnerability assessment.

After aggregating indicators, some studies apply a normalization aiming to ensure that the output from indicator aggregation
lies within defined intervals. These intervals should directly be suitable to communicate the extent to which an element at
risk is vulnerable. JRC (2008) pointed out that the choice of a normalization approach should be related to data properties
and underlying theoretical frameworks. Although there are several normalization techniques, most studies in flood
vulnerability assessment apply the minimum-maximum normalization (detailed descriptions of normalization methods can
be found in JRC, 2008). In the minimum-maximum normalization, index outputs are bound within a fixed range, commonly
between 0 (not vulnerable) to 1 (highly vulnerable). The minimum-maximum normalization can increase the range of small-
interval indicators or reduce the range of large-interval indicators. Hence, all indicators are allowed a proportionate effect on





the aggregated index. Other studies, however, do not use any form of index normalization, for example, in Akukwe and Ogbodo (2015), weights from PCA were directly aggregated to create an index without any normalization.

## 2.3 Challenges and gaps of vulnerability indicators and indices

### 2.3.1 Managing input data and pre-processing

Constructing indicators requires a systematic integration of different data sets (qualitative and quantitative, semi-quantitative), and different supporting knowledge (deductive, inductive or normative). For an index to be effectively operationalized in the vulnerability concept, underlying data has to be properly managed. Already UNDP (1992) highlighted that the quality of an index is dependent on the data it is developed from and suggested that careful pre-processing of underlying data should be carried out. For indicators to be policy-relevant, care is needed so that developed indices are not unreliable (Chen et al., 2012) or an empty communication tool (Günther, 2006). Data operations, such as filling in of missing data, scaling and normalization, influence the model output considerably and should be carried out using appropriate methods that fit the data type and indicator framework. Normally, in these data operations, several approaches exist and a selection of any approach will have implications on developed indices. For example, Chow et al. (2019) demonstrated the effect of three common methods for imputation of missing data and its implication on transformed data sets. In a study on vulnerability indicators, Tate (2012) asserted that data transformation highly influences index output and should receive the highest critical examination in developing indicators. Mosimann et al. (2018) presented a new way of data transformation for deducing a flood vulnerability model for household content and came to the same conclusion. Few points to examine and clearly communicate during the indicator development phase include questions such as "What relationship exists between indicators?", "What kind of scaling and normalization will be appropriate and why?", "How much will scaling and normalization influence the variable range in order to correctly transfer its theoretical meaning?", and "Which method was used for filling missing data and why?". In many studies, however, data transformation methods are either not mentioned or only briefly highlighted. Nonetheless, it is important for future research that a proper communication of data transformation methods is carried out to allow for guidance and discussions on harmonizing indicators across scales.

### 2.3.2 Need for improving performance test

Vulnerability indicators are used to analyse disaster risk. Thus, the framework of an index formulation should allow adaption to possible future changes. The accuracy of the constructed index should be checked by applying performance assessment or validation test. A performance test will allow a robust evaluation of underlying indicator framework and basic assumptions (Eddy et al., 2012) and the suitability of selected indicators with respect to the study aim (Birkmann, 2006). A performance test will help in developing sound indicators (Asadzadeh et al., 2017) since it will communicate index accuracy and success in achieving a redefined indicator aim. Success in achieving a defined indicator aim will foster transferability to data-scarce regions with similar characteristics in hazard and built environment. However, in physical vulnerability assessment, index performance or validation are rarely carried out. Few studies provide a qualitative description (e.g. level of agreement) as performance analysis based either on a comparison of a deduced index and observed damage data. (e.g. Godfrey et al., 2015; Sadeghi-Pouya et al., 2017) or visualise the spatial agreement on GIS map by presenting and comparing computed hotspots of the index and the observed damage data (e.g., Fernandez et al., 2016). A lack of performance test might likely be due to the scarcity of empirical data in many regions. A systematic linkage between the vulnerability index and a hazard impact parameter (e.g., building damage or monetary loss) can be a viable option to support performance check. Due to this linkage,





the output of vulnerability index will not only be an index expressing relative vulnerability to other buildings but will be extended to expressing a physical damage state which can be observed and documented.

### 2.3.3   Understanding sensitivity and uncertainty

Several approaches in selecting, weighting, scaling, normalizing and aggregating indicators exist, these consequently introduce different levels of subjectivity into the index output. Hence, it is important that different approaches of index 450 generation are checked against uncertainties and the sensitivity of the index. JRC (2008) and Tate (2012) stressed the need for an internal validation to assess the robustness of indices and check how each approach influence the index stability. Furthermore, such analysis can convey the extent of the index reliability and enhance transparency in indicators to guide future index development. For example, Tate (2012) carried out a sensitivity and uncertainty analysis for each step of indicator generation. One finding from Tate (2012) was that both, inductive and deductive approaches are sensitive to 455 indicator choice, and, as such, statistical properties of an indicator should be thoroughly considered in the selection phase. Another example of a comprehensive sensitivity and uncertainty analysis was carried out by Saisana et al. (2005) using the United Nations Technology Achievement Index (TAI). By contrast, no detailed uncertainty and sensitivity assessment has been carried out yet for physical vulnerability indicators. A study by Fernandez et al. (2016), however, has taken first steps to improve the understanding of the sensitivity of vulnerability index in physical vulnerability assessment. They used a GIS 460 approach to check the sensitivity of index performance to different aggregation methods. Results indicate that while additive methods (direct and weighted additive method) show similar results, an aggregation using cluster analysis exhibited better results in detecting hotspots when compared to recorded damage data. Hence, detailed sensitivity and uncertainty analysis are highly recommended for assessing physical vulnerability.

### 3 Overview of flood damage models

Flood damage models (or loss models) are used to show the relationship between the extent of building damage on one hand and hazard and building characteristics (damage influencing factors) on the other hand. The applications of these models are varied depending on user need. Flood damage models provide basis for decision making on cost-benefit analysis of mitigation measures (Thieken et al., 2005; Schröter et al., 2014), economic impact assessment (Jongman et al., 2012), for planning and implementing individual mitigation measures (Walliman et al., 2011) and for flood risk mapping (Schröter et 470 al., 2014). In addition, flood damage models provide a basis for estimating compensation costs for individuals and for loss comparison between regions. Additionally, loss estimates are useful for checking the utility of flood protection or mitigation measures between the past and present.

Prior to developing a damage model, it is important to define important aspects and functions of the model has to be defined. A decision has to be taken if the model is for damage grade evaluation (e.g., Ettinger et al., 2016), absolute or relative 475 monetary loss assessment (e.g., Thieken et al., 2008) or for both (e.g., Maiwald and Schwarz, 2015). The spatial scale (individual or aggregated buildings) and time scale (immediate or long time damage) has to be defined (Blong, 2003b). Some damage models however are applicable for assessing damage at individual and aggregated scales (e.g., Maiwald and Schwarz 2015). Damage models should also be flexible to allow integration of new damage patters over time (Blong, 2003b), such as shown by Maiwald and Schwarz (2015, 2019) when expanding the original five-category damage grade 480 scheme to a six-category scheme. In the next sections, we further assess specific features of flood damage models to evaluate their suitability or otherwise in a data-scarce region.



### 3.1 Damage grade model

Developing a flood damage model requires data or knowledge on building damage patterns resulting from hazard impact. Such building damage patterns that are repeatedly observed within a region can be categorized into a damage grade.

Building damage occurs in various extents from light to heavy damage or collapse, usually referred to as damage grades. Damage grades vary from non-structural to structural damage. Non-structural damage refers to damage that does not immediately affect the structural integrity of a building. Examples of non-structural damage by floods include moisture defects or light cracks on building finishes. Structural damage mostly occurs on load bearing elements of the building, for example, cracks or collapse of walls, beams, columns (Milanesi et al., 2018). One of the most detailed development of

damage grades was contained in the European Macroseismic Scale EMS-98 by Grunthal (1998). The scheme, developed by Grunthal (1998) for earthquakes, was later used as a basis to develop damage models for flood by Schwarz and Maiwald (2007). Grünthal (1993) recommended few guidelines for good practice in developing a damage grades, these include (i) checking a wide range of information sources and consider their value, (ii) focusing more on repetitive damage than on extreme damage pattern, and (iii) additionally considering undamaged buildings.

Table 4 presents the damage grade scheme developed by Schwarz and Maiwald (2007): it shows categories of damage states from water penetration to both structural and non-structural damage. Damage grades express damage patterns as categories on an ordinal scale whereby numbers are assigned to each damage pattern with higher numbers depicting higher degree of damage (see Table 4). Some damage grade models (e.g., Maiwald and Schwarz 2015) are applicable for assessing damage at individual and aggregated scales. As a recommendation for good practice, damage models should be flexible to allow

integration of new damage patters over time (Blong, 2003b). An example of such flexibility is demonstrated in Maiwald and Schwarz (2015, 2019) when expanding the original five-category damage grade scheme to a six-category scheme.

Although damage prediction models are widely used in several countries, difference in regional building types have limited the transferability of such models (Tarbotton et al., 2015; Englhardt et al., 2019). For example, studies by Maiwald and Schwarz (2015) show that empirically-deduced damage grades allow a transfer of the damage distribution and vulnerability

models in study areas with close agreements of the observed effects. However, a particular advantage of damage grades is that they are physically observable patterns and can be qualitatively described and used to compare impacts of hazards across spatial and temporal scales. This is particularly useful in many data-scarce regions where monetary loss data (e.g., insurance data) are unavailable and the consequence of hazards can only be quantified using observable damage patterns or grades.

### 3.2 Stage-damage curves

Stage-damage functions are a quantitative method for assessing the vulnerability of buildings, and they are widely used for assessing flood hazard risk where the number of affected buildings is large enough to deduce a reliable curve (Fuchs et al., 2019a). These functions are continuous curves relating the magnitude of a hazard process (X-axis) to the damage state of a building (Y-axis), usually expressed as degree of loss. A general requirement of stage-damage functions is the confined interval needed, meaning that the function should run through the (0,0) point because zero magnitude causes zero degree of

loss (Fuchs et al., 2019b). Moreover, the function should not exceed the limit of degree of loss 1, because a degree of loss higher than 1 is only possible if costs other than direct damage are included in the vulnerability assessment.

Stage-damage curves can be developed using empirical data or synthetic approach. The shape of the empirically derived stage-damage function depends on the population and spread of data related to buildings within the inundation area under





consideration as well as the type of function chosen. Individual buildings are represented as points on a XY axis system and

then the function that ensures the best fit may be chosen (Totschnig et al., 2011). Synthetic stage-damage curves are developed by utilizing expert-based what-if analysis to develop a relationship between flood damage with flood depth for specific building or land-use types. For example, experts can be are asked to estimate the pattern of building damage that will be expected if a specific building type (e.g., masonry building) is inundated with a certain flood depth (e.g., 1 m). Synthetic methods can be developed independently (e.g., Penning-Rowsell et al., 2006; Naumann et al., 2009) or supported

by empirical data (e.g., NRE, 2000).

Despite the wide usage of stage-damage curves, several studies have highlighted the high uncertainty in these curves particularly because they only consider flood depth as the only damage influencing parameter (e.g., Merz et al., 2004, 2013; Vogel et al., 2012; Pistrika et al., 2014; Schröter et al., 2014; Wagenaar et al., 2017; Fuchs et al., 2019b). These studies have demonstrated that damage-influencing variables are not only related to water depth but also to other flood characteristics

(e.g., velocity, duration) and building characteristics (construction type, quality, material). For instance, Merz et al. (2004) demonstrated the poor explanatory power of flood depth in explaining the variance in a damage data set. However, the main advantage of the synthetic stage-damage approach is that it can be developed in the absence of empirical data using the what-if analysis. This is particularly useful for data-scarce regions especially if it can be used within an approach can considers multiple damage-influencing variables.

**3.3. Multivariate approaches**

Multivariate approaches utilize empirical data to develop a relationship between damage on one hand and flood and building characteristics on the other hand. Such empirical data can be collected from insurance companies (e.g., Chow et al., 2019), through field surveys (e.g., Ettinger et al., 2016), or by telephone interviews (Thieken et al., 2005; Schwarz and Maiwald, 2008; Maiwald and Schwarz, 2015). Empirical data can also be collected using social media accounts like Twitter as recently

demonstrated by Cervone et al. (2016). Several statistical methods employed in literature for developing multivariate methods include Bayesian network by Vogel et al. (2012) , logistic regression by Ettinger et al. (2016), bagging decision trees by Merz et al. (2013) and Wagenaar et al., 2017, and logit-linear regression, double generalized linear models and random generalized linear model by Chow et al. (2019). Multivariate models are becoming common since, by including multiple damage-influencing variables, they offer a more comprehensive approach compared to the stage-damage curves.

Schröter et al. (2014) evaluated the usefulness of flood damage models and showed that models that consider higher number of damage influencing variables demonstrated superiority in predictive power both spatially (transfer to other regions) and temporally (different flood events). Multivariate functions have been shown to explain better the variability in damage data (Merz et al., 2004) and reduce uncertainty in flood damage prediction (Schröter et al. 2014). Table 3 represents a short overview of multivariate models developed for floods. Although there are many of such models in literature, selected studies

are used to show the wide variety of statistical methods employed, damage-influencing variables considered, scale of application, data source and performance test. The majority of multivariate models use mainly flood and building characteristics in their models. Some additionally consider building exposure parameters to include the characteristics of the building surrounding. For example, Maiwald and Schwarz (2015) included building location relative to other buildings and to the flood source as part of the damage model. Although it is very important for damage models to capture damage

influencing variables so as to improve its accuracy, having too many variables might lead to model complexity. Consequently, Walliman et al. (2011) pointed that damage models must strike a balance between accuracy and complexity



A drawback of the multivariate method is the empirical data requirement for deriving and validating such models. Such data require intensive time and human effort to collect (in a data-rich region) or are limited (or unavailable) in data-scarce regions. Nonetheless, a special similarity between the multivariate method and vulnerability indicators is that they both
integrate multiple damage-influencing variables. Since the vulnerability indicators can be developed using expert knowledge, we integrate it into the proposed new framework.

## 4    The need for linking indicators and damage grades

Using a synthetic what-if analysis to link damage grades (representing repeatedly observed damage patterns) and the vulnerability indicator (capturing important damage-influencing variables within a region) can produce a vulnerability model
that is simplistic and comprehensive for use in data-scarce regions. In general, the linkage of these approaches is a promising pathway taking advantage of the strengths of the two methods while limiting their individual weaknesses. Vulnerability indicators have been identified to pose challenges if transferred between different regions or used to compare the vulnerabilities between regions. In addition, current flood damage models are identified to be either data intensive (multivariate methods) or do not consider other damage-influencing variables (stage-damage curves). However, an
integration of damage grades with the vulnerability indicator can provide a suitable model that overcomes these challenges.

To demonstrate the value of this linkage, we use a combination of observed flood damage data, a hypothetical vulnerability index for two regions A and B, and two damage models for predicting damage grade. The observed damage data (see Fig. 2) was documented from a field survey conducted after the 2017 flood event in Suleja and Tafa areas in Nigeria. The flood event was caused by prolonged rainfall for about 12 hours from 8[th] to 9[th] July 2017, which resulted in loss of lives and
damage to hundreds of buildings and infrastructure (Adeleye et al., 2019). A field study was conducted in March 2018 during which affected household owners were interviewed and data on damage sustained by affected buildings were documented. Out of the documented damage cases, we use three buildings to illustrate the potential draw back in using only the vulnerability index approach and the added value of the suggested linkage with damage grades. The three buildings in Figure 2 are constructed from sandcrete block (buildings i, ii) and clay bricks (building iii). The buildings have different
damage patterns ranging from moisture defects on walls resulting in peeling-off of plaster material and slight cracks (e.g., building i), partial collapse of supporting wall (e.g., building ii) and complete collapse (e.g., building iii). A hypothetical vulnerability index is considered for the two regions A and B (see Fig. 2).  In the two regions, hypothetical vulnerability indicators were assigned as main damage-influencing parameters in the regions. Indicators for region A include building material, building condition, distance to channel and flood depth. Indicators for region B include building age, building
quality, sheltering effect and flood depth. Vulnerability indices for regions A and B both express relative vulnerability from 0 (low vulnerability) to 1 (high vulnerability). Hypothetical vulnerability indices after aggregating identified indicators are given in Figure 2. We further consider two damage grades developed by Maiwald and Schwarz (2015) for Germany and by Ettinger et al. (2016) for Peru. We use identified damage patterns on the buildings from the field study to assign a damage grade to each building. From Fig. 2, we see that although we can use the developed index to identify which building is
highly or moderately vulnerable within a region, we cannot compare the indices between different regions because they contain aggregated information from different parameters. However, in the case of damage grades, although they were developed in two different regions, qualitative descriptions of the damage grades can be used to assign damage grade classes for the identified damage patterns in buildings i, ii, iii (Fig. 2). Using such comparability that comes from the consistency of hazard consequence, a model that predicts damage can allow us to compare hazard impacts across spatial and temporal
scales.





A combination between vulnerability indicators and damage grades has a number of advantages for data-scarce areas. These include;

    i.    Employing the synthetic method to link damage grades and hazards parameters, we can overcome high data requirement of the empirical method. Consequently, the linkage will capture building and hazard characteristics through the vulnerability indicators. This will help overcome limitation of the synthetic stage-damage method since multiple damage influencing parameters can be integrated

    ii.    The linkage will allow us to compare consequences of flood hazards in different data-scarce regions using the damage grades and at the same time considering regional differences using the vulnerability indicators. This will foster transferability since we can have common observable damage features resulting from flood impact.

    iii.    Since damage grades are physically observable features, the linkage with vulnerability indicators will allow us to carry out performance check on the effectiveness and robustness of vulnerability index.

## 5  Conceptual framework

### 5.1  Background for operationalizing the new framework

We present a new framework that aims to link vulnerability indicators on one hand and damage grades on the other hand to make use of their individual strengths. The vulnerability indicators are used to capture damage-influencing variables which include flood hazard characteristics and the elements of the built environment and its surrounding as shown in Fig. 3. The damage grades represent physical consequences of hazard impacts on a building which depends on both hazard and building characteristics.

The vulnerability indicator considers two major damage influencing variables; action (impact) and resistance variables (Thieken et al., 2005; Schwarz and Maiwald, 2007). Action (or impact) parameters relate to the flood parameters comprising of hazard frequencies and magnitudes. Resistance parameters are related to the predisposition of the building to suffer damage, either permanently (e.g., building material) or temporarily (e.g. measures for flood preparedness) (Thieken et al., 2005). Resistance parameters comprise of elements of the building and its surrounding, which are divided into susceptibility, exposure, and local protection variables. Vulnerability is seen as the degree to which an exposed building will experience damage from flood hazards under certain conditions of susceptibility and resilience (Balica et al., 2009). Exposure refers to the extent to which a building is spatially or temporarily affected by a flood event (Birkmann et al., 2013). Exposure parameters include features of the natural and built environment that either increase or decrease the impact of floods on buildings, such as topography and distance to the flood source. Susceptibility refers to the disposition of a building to be damaged by a flood event (modified after Birkmann et al., 2013). Susceptibility parameters relate specifically to the structural characteristics of the building at risk, neglecting any effects of local protection measures which may provide flood protection. Local protection in this study refers to deliberate or non-deliberate measures that are put in place and can reduce the impact of the floods on a building. These measures can be directly included into the building structure e.g. elevation of entrance door, located in the immediate surrounding of a building. While many local structural protection measures may not be primarily constructed as a protection mechanism against floods, they reduce the impact of floods on a building. In the context of this framework, a fencing wall will be an example of a local protection measure.

Damage grades represent classes of regularly observed damage patterns (Maiwald and Schwarz 2015). Generally, there are a wide range of damage patterns to describe how buildings respond to flood impact. Including all these patterns however will


lead to unnecessarily difficult and complex damage prediction models. Nonetheless, damage grades should be detailed enough to capture predominantly observed patterns of damage within a region. Damage grades serve as a compromise

between comprehensiveness and simplicity (Blong 2003b). Some characteristics of a damage grades given by Blong (2003b) include simplicity, clarity, reliability, robustness, and spatial suitability. Damage grade should not only consider physical effects of damage but also the quantity of buildings that show such effects (Grünthal et al., 1998; Maiwald and Schwarz, 2015).

### 5.2 Operationalizing the framework

In order to operationalize the new framework, three basic phases are proposed; (i) developing a vulnerability index (ii) developing a damage grade classification (iii) linking vulnerability index and damage grades.

#### 5.2.1 Phase 1: Developing a vulnerability index

We develop here a vulnerability index aimed at identifying damage influencing variables. As a result, we structure indicators into impact (action) and resistance parameters as seen in Fig. 3 (phase 1). Implementation of the method will depend on data

availability, therefore we propose both a data-driven and an expert-based approach. Both approaches can be supported by literature review. Resistance parameters are categorized into separate components (e.g. exposure, susceptibility, resilience) in order to allows an evaluation of how different components contribute to damage. Application of the method is aimed at micro-scale level, however, it can be applied at meso or macro scale if data are available. The selection, weighting and aggregation of indicators are similar to the procedure discussed in section 2.2.

*Indicator selection*

Selection of indicators is carried out through a data-driven approach or based on expert knowledge. Data driven approaches utilize weights from the PCA to identify important damage influencing parameters. Where such empirical data is not available, as is the case in many developing countries, expert knowledge is utilized through conducting standardized interviews using questionnaires. If there are studies conducted in regions with comparable building and hazard

characteristics, identified damage-influencing parameters can be used to provide basis for the pre-selection of indicators. Such pre-selection will guide final selection of indicators in both the data-driven and expert approach. Considering regional comparability is highly important as demonstrated in a study by Cammerer et al. (2013); the study showed that damage functions performed considerably well when tested in regions with similar building and flood characteristics. Cammerer et al. (2013) also showed the relatively poor performance of flood damage models from regions with different building and

hazard properties.

*Indicator weighting*

Indicator weighting is carried out using either an expert-based or a data-driven approach. For the expert weighting, selected group of experts are asked, using questionnaires or through interviews, to weight how each selected indicator influences damage. The weighting is carried out using a scale of influence measure as shown in Table 5 based on Saaty (1980).

Although the table by Saaty (1980) is used to make a pair-wise comparison between two parameters, we slightly modified it so as to be used in weighting a parameter with respect to flood damage. The scale (Table 5) will help to bring consistency and comparability in weighting when using the framework. Using the scale, experts can assign a certain influence for each parameter. For each indicator, a mean value of the assigned weights from all experts is calculated. The mean weight for each





indicator is used to further determine parameter inclusion in the aggregation step. For the data-driven approach, a PCA is
employed to weight all selected parameters. PCAs have been employed in previous studies both with (e.g., Thieken et al.,
2005; Kreibich et al., 2010) and without empirical flood damage data (e.g., Thouret et al., 2014; Akukwe and Ogbodo,
2015). In both cases, factor loadings from PCA determine the extent of influence a parameter has on damage.

In the two methods outlined, weights extracted from both the PCA (factor loadings) and expert based (mean weights) can be
used to determine if a parameter is included for the aggregation or not. For example, a mean weight of 2 from Table 5 will
infer that most experts consider the parameter to have only a slight effect on damage. Similar methods for reducing initially
selected parameters are used in empirical damage models. For example, initially selected parameters where eliminated
during the logistic regression due to low influence on damage by Ettinger et al. (2016). In adopting the procedure, decision
has to be made on a threshold for parameter inclusion in the aggregation step. The threshold will depend on specific function
(e.g., level of accuracy) or aim (e.g., identifying major damage influencing parameters) of the study.

*Indicator aggregation*

A normalized weighted additive method is used for aggregating indicators. As shown in Fig. 3 (phase 1), selected parameters
for exposure are aggregated to derive a Building Exposure Index (BEI). The BEI is a measure of the extent to which a
building is likely to be damaged as a result of (i) the spatial location relative to the flood source and (ii) surrounding
buildings. Variables of susceptibility and local protection are aggregated to derive a Building Predisposition Index (BPI).
The BPI provides a measure of the extent to which a building is likely to be damaged based the building characteristics and
available protection measures. Both BEI and BPI are aggregated to derive a Building Resistance Index (BRI). The BRI
measures expected behaviour of a building considering its predisposition and exposure. The BRI is used to classify buildings
into different resistance classes (e.g., low, moderate and high). Such classifications of buildings into vulnerability categories
have been shown to facilitate a better understanding of the distribution of a damage data (Schwarz and Maiwald, 2008).
Elements within the same vulnerability class are expected to experience similar damage when impacted by the same hazard
level. Furthermore, we utilize an additive model to aggregate flood hazard parameters (e.g. depth, duration) in order to
derive a Building Impact Index (BII). The BII is used to express the combined effect of hazard variables on a building
structure. The BII is computed using collected data after a flood event or by utilizing flood scenario modelling to extract
hazard parameters (Mark et al., 2019).

### 5.2.2 Phase 2: Developing damage grades

We adopt a slightly modified procedure outlined in Naumann et al. (2009) to develop damage grades. Figure 3 (phase 2)
shows the systematic steps for developing the grades using an expert-based approach. Main aim of this step is to identify
commonly observed damage patterns within a region and categorize them into classes. As such, basic outputs of this phase
are classes of different damage grades. We present an overview of a synthetic method for developing a damage model as
described by Naumann et al. (2009). The necessary steps include:

    i.    Identification of buildings with representative: An important first step for developing a flood damage model is to
assess building types within region and select representatives (Walliman et al., 2011; Maiwald and Schwarz, 2015).
The assessment of building types can be carried out through field surveys or remote sensing. Where a large-scale
building assessment is required a method conceptualized by Blanco-Vogt and Schanze (2014) for semi-automatic
extraction and classification of buildings using a high-resolution remote sensing data can be applied. Naumann et al.
(2009) noted that attributes that are used for classifying buildings could include: period of construction and original



use, characteristic formation of buildings, spatial pattern and geometry and construction details. From each BRI class, a representative building will be selected. Suitably, these representatives can be selected from different building types.

ii. Identification of regional damage categories: Flood damage to buildings can be generally categorized into three major parts according to these include water penetration damage (moisture), chemical damage (pollution and contamination) and structural damage (Schwarz and Maiwald, 2007; Walliman et al., 2011). These three general damage categories can serve as basis for developing damage classification in regions where such damage assessment was not previously carried out. Structural damage should be further examined to develop sub-categories

of damage patterns within a study region. For each BRI representative building, different patterns of structural damage are identified by a group of experts. Patterns which are repeatedly observed are indications of a damage grade category. Decision has to be made on how many damage grades to considered between a minimum damage grade of water penetration and a maximum damage grade of total collapse (see for comparison Table 4). Where studies have been conducted for a region with comparable building types, damage patterns can be used as a guide.

Other information sources such as news reports, social media (videos and images) can be utilized in identifying damage patters. Identified damage categories are assigned to an ordinal scale to show a minimum damage (usually water penetration) to maximum damage (complete collapse or washing away of a building).

### 5.2.3 Phase 3: developing a relationship between vulnerability index and damage grades

Several methods have been used in relating damage grades to damage influencing variables. Selecting a particular method is
likely depending on data availability and knowledge on damage mechanisms (Merz et al., 2010). With a focus on data-scarce regions, we outline steps using an expert-based approach to link damage grades and damage influencing variables. Blong (2003b) asserted that in absence of empirical data for developing flood damage models, a link between building damage and significant parameters contributing to damage (indicators) can be established. Furthermore, the approach can be enhanced by developing an aggregated value (or index) to qualify damage expressed in terms of damage grades. Here, the basic idea is to
use expert knowledge with artificial inundation scenarios to construct a flood damage model as applied in the synthetic method. We use expert knowledge to define probable damage grades for selected intervals of BII. Each BRI class will have a different model. Figure 3 (phase 3) shows an idealized curve depicting the relationship between damage grades, BII and BRI. Methodical steps for developing the link between damage grades with the BRI and BII are modified after a similar approach used in Maiwald and Schwarz (2015). Instead of empirical data however, a synthetic "what-if" analysis is adopted
for data-scarce regions. The following steps are illustrated for linking damage grades with the developed indices.

i. Classify BRI into five building resistance classes (very low, low, moderate, good, very good)
ii. Select one building type as representative for each BRI class. Ideally, the representative building should include different building materials or construction types within the BRI class.
iii. Develop suitable intervals for the BII, such as flood depths in steps of 0.5 meters.
iv. For each defined interval of BII, local experts are asked to estimate the expected damage for each BRI class. Instead of estimating one single damage grade, however, experts should provide three possible damage grades for each BII interval. The possible damage grades should include (i) most-probable damage grade (ii) lower-probable damage grade (iii) higher-probable damage grade. The three grades are to represent the uncertainty bounds of the actual damage. For example, if a representative building type (one-storey sandcrete block
building) is selected from BRI category "low resistance", experts will estimate for each BII interval (e.g., 1 m



water depth) the damage to be expected: (i) most-probable: slight cracks on supporting walls, (ii) lower-probable damage: only water penetration, (iii) higher-probable damage: heavy cracks on supporting walls. The need for using three probable damage grades is to capture uncertainty range associated with each BII for given building and surrounding characteristics (BRI class).

v.     For each BRI class, a separate a suitable curve is used to link each BII intervals and most probable damage grade. Uncertainty bounds are shown on each curve to communicate possible scatter range. Figure 3 (phase 3) shows an idealized example.

## 6    Conclusion

With increasing magnitudes and frequencies of floods, assessing the vulnerability of communities exposed is crucial for
reducing risk. The success of risk reduction methods is even more critical for developing countries due to their limited capacity to adapt to flood events. Vulnerability assessment incorporates identification of major drivers of damage and loss of exposed buildings. Developing a model to estimate future damage under probable scenarios may serve as a first step in overall risk reduction. So far, scarcity of empirical data has limited such efforts in many regions. In this study, we proposed a conceptual framework for linking the vulnerability indicator approach to damage grades using expert knowledge in order to
develop a flood damage model for data-scarce regions. Combining such methods has been identified as a useful way to enhance the utility and robustness of vulnerability assessments while limiting drawbacks resulting from the use of only one method. The proposed framework focuses on enhancing regional adaptability of vulnerability assessment methods and fostering model transfer between different data-scarce regions. Three phases were adopted to develop the framework, (i) developing a vulnerability index, (ii) identifying predominant damage grades or patterns, and (iii) expert what-if analysis to
link identified damage grades to flood characteristics for each class of building and surrounding characteristics.

In developing the vulnerability index, we considered hazard variables (BII) on one hand and the variables relating to the characteristics of a building and its surrounding (BRI) on the other hand . The BRI aggregates information on a building's exposure, susceptibility and local protection and communicates the resistance the building can offer relative to other buildings assuming similar hazard magnitudes. The classification of the BRI here is not building-type based (e.g., Maiwald
and Schwarz 2015) but based on aggregated information on exposure, susceptibility and local protection such as property-level flood risk adaptation measures. Although Maiwald and Schwarz (2015) recommended a building-type vulnerability classification, it is, however, not feasible in many regions due to (i) low variability in building types, and (ii) high variability in building quality within the same building type. In some countries of Africa such as Nigeria for example, sandcrete block and clay buildings make up over 90 percent of the building types (NBS, 2010) but those buildings show variable
susceptibility to flood hazards due to a wide variability in building quality (FGN, 2013). Recently, Englhardt et al. (2019) made similar observations in Ethiopia, pointing out that in many rural areas even a low flood depth can result in high damage due to low building quality. In such areas, where a building type vulnerability classification will be unsuitable, we recommend a generic vulnerability classification (e.g., low, moderate, high resistance) such that buildings are classified into categories based on their expected resistance to flood with respect to identified damage-influencing variable. A systematic
documentation of building damage grades by local experts is required for the framework allowing the categorization of frequently observed damage patterns (e.g., moisture defects, cracks on supporting elements, partial collapse, complete collapse). As the framework is not case-study sensitive, damage categories from other studies can provide useful basis for categorizing probable damage classes.



Furthermore, a what-if analysis is used to assign identified damage grades to each interval of the BII for each class (e.g., low,
moderate, high) of the BRI. Since expert knowledge are subjective, the use of three damage states (most probable, lower
probable and higher probable) for each BII interval is important for capturing the range within which the damage for a BRI
class will lie. A similar approach has been used by Maiwald and Schwarz (2015) for the assessment of building-type
vulnerability classes. The BII can be adapted to regional requirements such as, for example, in cases where unburnt clay
buildings are predominant, a combination of flood depth and duration might be used for BII intervals since long-duration
floods tends to weaken such building types. The presented framework is flexible so that correlations between BII, BRI and
damage grades can be continuously updated with new data within a region or from a region with comparable building and
hazard characteristics. In cases few empirical data is available, even in limited quantity, it should be used to support the
assignment of damage grades. The framework is flexible and the vulnerability indicators and damage grades can be updated
when new post-flood data becomes available. Consequently, temporal changes in damage drivers can be integrated to adapt
the model.

The proposed framework will foster model adaptability to regional conditions and allow transferability between comparable
regions. The framework will also extend the application of vulnerability indices towards a value which can be used to predict
flood damage. The BRI classification offers information on buildings with low resistance to floods. Such information is
important for emergency and mitigation planning. Application of the framework for damage prediction will enable the
economic evaluation of disaster scenarios to support planning on disaster risk in data-scarce regions. Limitations of the
method include the subjectivity of the expert-based approach. The potential in utilizing additional data sources such as social
media offers a good opportunity for damage data collection to support expert assessment, and may also be used to reduce the
subjectivity of initial expert assessments.

**Author contributions**

MM designed the study and was responsible for data collection, analysis and literature review, MK and SF supported with
literature review and analysis, all authors were jointly involved in manuscript preparation and editing.

**Competing interests**

Margreth Keiler and Sven Fuchs are members of the Editorial Board of Natural Hazards and Earth System Sciences.

**Acknowledgement**

This study is carried out within the framework of a PhD scholarship funded by the Swiss Government Excellence
Scholarships for Foreign Scholars (ESKAS). Thanks to Candace Chow for useful inputs in formulating Figure 2.

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



**Tables**

Table 1: Overview of common elements for framing the vulnerability indicator approach for flood hazards, indicating the hazard type and vulnerability dimensions, the implementation in the risk cycle, the scale, and the index output as well as the data source.

| Author | Hazard type | Region of application | Aim of the assessment | Vulnerability dimension | Implementation in risk cycle | Scale | Index output | Data source (vulnerability of buildings) |
|---|---|---|---|---|---|---|---|---|
| Papathoma et al. (2003) | Tsunami | Gulf of Corinth, Greece | Assessing the vulnerability of coastal areas to tsunami | Physical, economic and social | Preparedness | Local-scale (individual buildings) | Building and human vulnerability index | Field survey |
| Dominey-Howes and Papathoma (2007) | Tsunami | Maldives, India | Checking the performance of PTVA | Physical, economic, social and environmental | Preparedness | Local-scale (individual buildings) | Building and human vulnerability index | Field survey |
| Balica et al. (2009) | River flood | Timisoara, Romania; Mannheim, Germany; Phnom Penh, Cambodia | Assessing the conditions influencing flood damage at various spatial scales | Physical, economic, social and environmental | Preparedness | Meso-scale (regional) | Flood vulnerability index | - |
| Kienberger et al. (2009) | River flood | Salzach catchment, Austria | Identification of hotspots | Physical, economic and social | Mitigation and preparedness | Meso-scale (regional) | Vulnerability index | Government agency |
| Dall'Osso et al. (2009) | Tsunami | Sydney, Australia | Assessing vulnerability of buildings to tsunami and assess use of PTVA | Physical | Mitigation and preparedness | Local-scale (individual buildings) | Relative vulnerability index | Field survey |
| Müller et al. (2011) | (Urban) flood | Peñalolèn and La Reina Municipalities, Santiago de Chile | Empirical investigation of vulnerability towards flood | Physical and social | Mitigation | Local-scale (entire building blocks) | Vulnerability index (adapted after Haki et al., 2004) | Census data, field survey and satellite data |



| Kappes et al. (2012) | River flood, flash flood (among others) | Faucon municipality, Barcelonnette basin, France | Assessing the hazard-specific physical vulnerability of buildings towards multi-hazard | Physical, social and environmental | Mitigation and preparedness | Local-scale (individual building) | Relative vulnerability index | Research agency and aerial-photo-interpretation |
|---|---|---|---|---|---|---|---|---|
| Balica et al. (2012) | Coastal flood | Buenos Aires, Argentina; Calcutta, India; Casablanca, Morocco; Dhaka, Bangladesh; Manila, Philippines; Marseille, France; Osaka, Japan; Shanghai, China; Rotterdam, The Netherlands | Developing a coastal city flood vulnerability index | Physical, social, economic and administrative | Preparedness | Meso-scale (regional) | Coastal city flood vulnerability index | Government agencies and data available online |
| Blanco-Vogt and Schanze (2014) | River flood | Magangué, Columbia | Assessing physical flood susceptibility on a large scale | Physical | Recovery, mitigation, and preparedness | Local-scale(individual buildings) | Function relating susceptible material volume and water depth | Very high resolution spectral and elevation data and field survey |
| Thouret et al. (2014) | Flash flood | Arequipa, Peru | Assessing vulnerability | Physical and environmental | Mitigation | Local-scale (entire building blocks) | Vulnerability index | Field survey |
| Bagdanavičiute et al. (2015) | Coastal flood | Coast of Lithuania | Assessing coastal vulnerability | Physical | Mitigation | Meso-scale (regional) | Coastal vulnerability index | Field survey |
| Behanzin et al. (2015) | River flood | Niger River Valley, Bénin | Assess vulnerability and risk | Physical, economic, social and environmental | Mitigation and preparedness | Meso-scale (community) | Vulnerability and risk index | Field survey, other agencies |



| Godfrey et al. (2015) | River and flash flood, slow moving landslide, debris flow | Nehoiu City, Buzău County, Romania | Assessing physical vulnerability of buildings to hydro-meteorological hazards in data-scarce regions | Physical | Mitigation and preparedness | Local-scale (individual buildings) | Vulnerability index | Field survey and orthophoto interpretation |
|---|---|---|---|---|---|---|---|---|
| Akukwe and Ogbodo (2015) | River and coastal flood | Port Harcourt, Nigeria | Showing spatial variations in vulnerability | Physical, economic and social | Mitigation and preparedness | Meso-scale (regional) | Vulnerability Index (adapted after Deressa et al., 2008) | Field survey, survey and map measurements |
| Fernandez et al. (2016) | River flood | Vila Nova de Gaia, Northern Portugal | Providing an automated framework for classifying vulnerability of neighborhoods | Physical, economic, social and environmental | Preparedness | Local-scale (neighborhood) | Flood Vulnerability Index and | Government agency |
| Ntajal et al. (2016) | River flood | Mono River Basin, Togo | Assessing and mapping vulnerable communities | Physical, economic, social and environmental | Mitigation and preparedness | Meso-scale (community) | Index for exposure, susceptibility, capacity, and vulnerability | Field survey, other agencies |
| Krellenberg and Welz (2017) | Flood (urban) | Metropolitan area of Santiago de Chile | Assessing urban vulnerability | Physical, economic, social and environmental | Mitigation | Local-scale (building block) | Vulnerability Index | Field survey, government agency and satellite imagery |
| Sadeghi-Pouya et al. (2017) | River flood | Mazandaran, Iran | Assessing vulnerability | Physical, economic, social and environmental | Mitigation and preparedness | Local-scale (building block) | Relative vulnerability index | Field survey and government agency |
| Carlier et al. (2018) | River flood | Upper Guil catchment, southern French Alps | Assessing the physical and socio-economic consequence of hazards on elements at risk | Physical and social | Mitigation | Local-scale (individual buildings) | Potential damage index, potential consequence index | Government agency, field survey, and aerial imagery |
| Yankson et al, (2017) | Coastal flood | Accra, Ghana | Understanding flood risk in coastal communities | Physical and Social | Mitigation | Meso-scale (community) | Impact index vulnerability index | Field survey |



| Percival et al. (2018) | Coastal flood | Portsmouth, United Kingdom | Assessing risk from diurnal floods | Physical, environmental, social, economic | Mitigation | Local-scale (neighborhood) | Coastal flood vulnerability Index, Coastal flood hazard Index, Coastal flood risk index | Census data |
|---|---|---|---|---|---|---|---|---|
| Papathoma-Köhle et al. (2019) | Tsunami | Apulia, Italy | Assessing vulnerability from tsunami hazards to the built environment | Physical vulnerability | Mitigation and preparedness | Local-scale (neighborhood) | Building vulnerability index | Field survey |





Table 2: Overview of approaches and parameters to deduce building vulnerability indicators for flood hazards, including methods used for variable selection, weighting, and aggregation,
and providing the parameters needed for assessing building vulnerability.

| Author(s) | Variable selection | | Variable weighting | | Vulnerability aggregation | Parameters considered (pertaining to building vulnerability) |
|---|---|---|---|---|---|---|
| | Preliminary | Final (used in model or equation) | Approach | Consideration for scoring/weighting | | |
| Balica et al. (2009) | Literature | Experts | No weights | Conditions that induce flood damage | Direct additive method | Flood depth, duration, velocity and return period, proximity to the river, land use, topography (slope), building codes |
| Kienberger et al. (2009) | Experts | Experts (Delphi approach) | AHP | Relative importance and contribution to the vulnerability of people | Weighted additive method | Buildings, infrastructure (transportation system), land cover |
| Müller et al. (2011) | Literature, field survey, experts | Experts | Expert knowledge, household surveys | Relevance of selected variables with respect to flood risk | Weighted additive method | Material for roof, walls and floor, position of building in relative to the street level, proportion of green spaces per building block, flood protection measures |
| Kappes et al. (2012) | Literature | Experts | Expert appraisals | Ability of the building to withstand the impact of the process | Weighted additive method | Building type, building use, building condition (using age and maintenance), building material, number of floors, row towards the river, trees towards the river |
| Thouret et al. (2014) | Literature, experts | Experts | Equal weights, experts, PCA | Weakness relative to a given hazard magnitude | Direct additive method | Heterogeneity of city block (using building size and use), building type (height and number of story, construction material, roof type and building condition), shape of city block, building density |
| Blanco-Vogt and Schanze (2014) | Literature, experts | Literature, experts | Literature, experts | General resistance characteristics after flooding (biological, chemical and material) | Weighted additive method | Building height, size, elongatedness (height/width ratio), building compactness, adjacency, roof, slabs, external fenestration, external wall, floor |
| Godfrey et al. (2015) | Literature, experts | Experts | AHP | Based on hazard impact | Normalized weighted linear combination | Floor height, number of floors, structural type, building size, wall material, presence of basement, number of openings, quality of construction, building maintenance, protection wall |



| Behanzin et al. (2015) | Literature | Experts | Equal (no) weights | - | Direct additive method | Building material, roof material, floor material, land cover around building |
|---|---|---|---|---|---|---|
| Akukwe and Ogbodo (2015) | Literature | PCA | PCA | Significance in explaining the variance in indicator data set | Weighted additive method | Building material, proximity to water, flood depth, flood frequency |
| Fernandez et al. (2016) | Literature | PCA | No weights and PCA | Significance in explaining the variance in data set | Direct and weighted additive method | Building density, number of floors, construction period, building material |
| Ntajal et al. (2016) | Literature, experts | Experts | Equal (no) weights | - | Direct additive method | Distance to river, flood depth, flood duration, building and roof material, land cover (area around building) |
| Krellenberg and Welz (2017) | Literature, experts | Experts | Equal (no) weights | Probability to be exposed under certain socio-environmental conditions | Direct additive method | Building quality, building structure, protection wall, trees in foreyard, roof form, land cover, housing condition |
| Sadeghi-Pouya et al. (2017) | Literature, experts | Experts | Experts (scoring) | Variable influence on vulnerability | Direct additive method | Building quality (material), building age, number of floors, land use |
| Carlier et al. (2018) | Literature | Literature, experts | Experts | Total consequence of a natural hazard on an element at risk | Weighted additive method | Building material, building condition, building age, building function, opening in hazard direction, building in area affected by flood (recurrence interval), land cover |



Table 3: Overview on empirically developed (multivariate) flood damage models, indicating the data source, the significance of variables, the scale of application, the parameters needed for developing the vulnerability function, and, where appropriate, the validation or performance test.

| Author | Case study/ Region of Application | Study aim | Data source for physical vulnerability indicators | Variable significance | Scale of application | Sample size | Parameters considered for developing the vulnerability function (Pertaining to physical - building vulnerability) | Validation or performance test |
|---|---|---|---|---|---|---|---|---|
| Thieken et al. (2005) | Germany | Investigation of flood damage and influencing factors | Computer aided phone interviews | Principal Component Analysis (PCA) and quantile classification | Local-scale (individual building) | 1697 | Flood depth, duration and velocity, contamination, precautionary measures, building type, building size, building quality | |
| Thieken et al. (2008) | Germany | Develop a model for flood loss (direct monetary) estimation for private sector | Computer aided phone interviews | (Multi)factor analysis | Local-scale (individual building) and meso-scale (regional) | 1697 | Flood depth, building type (Occupancy), Building quality, precaution, contamination | Using a different data set |
| Vogel et al. (2012) | Germany | Flood damage assessment of residential buildings | Computer aided phone interviews | Bayesian network | Local scale (individual building) | 1135 | Flood depth, velocity and duration, contamination, return period, precautionary measures, building type (occupancy), building size (floor space), building value, number of flats in building | Using subset of training data (bootstrap samples) |
| Merz et al. (2013) | Germany | develop tree based damage prediction models and compare their performance to established models | Computer aided phone interviews | Regression trees and bagging decision trees | Local-scale (individual building) | 1103 | Flood depth, velocity and duration, contamination, return period, precausionary measures, building type (occupancy), building size (floor space), building quality | Using subset of training data |
| Spekkers et al. (2014) | Netherland | Investigate damage influencing factors and their relationships with rainfall-related damage | Insurance data data and data from government agencies | Poisson (decision) trees | Meso-scale (district) | | Rainfall (intensity, volume and duration) related variables, building age, ground floor area, real estate value | Using subset of training data |



| | | | | | | | | |
|---|---|---|---|---|---|---|---|---|
| Ettinger et al. (2016) | Peru | Analysis of building vulnerability | Field Survey and analysis of high spatial resolution images | Logistic regression | Local-scale (individual buildings) | 898 | Distance from channel, distance from bridge, shape of city block, structural building type (material), building footprint | Using subset of training data |
| Maiwald and Schwarz (2015) | Germany | Develop engineering vulnerability-oriented for damage and loss prediction | Questionnaire survey, computer aided phone interviews, evaluation of damage reports, flood simulation | Tangent hyperbolic (damage grade) and an exponential function (relative loss) | Local-scale (individual building) and Meso-scale (regional) | | Flood depth and velocity, specific energy (flood depth, velocity and acceleration due to gravity), Building type, presence of basement, building location with respect to flow direction, number of stories | Using a different data set |
| Wagenaar et al. (2017) | Netherlands | Prediction of absolute (monetary value for content and structural) flood damage | Experts, flood simulation, Cadaster information | Bagging trees | Local-scale (individual buildings) | 4398 | Damage data (content and structural), flood depth, duration and velocity, building footprint, return period, building age, building area (footprint, living), basement, detached house | Using a 'withheld' part of the data set |




Table 4: Damage grades developed by Maiwald and Schwarz (2007) showing structural and non-structural damage to buildings. For each damage grade class, a description and a graphical representation are shown. The grey colour in the graphical representation indicates flood depth.

| Damage grade class | Damage | | Description | Graphical representation |
|---|---|---|---|---|
| | *Structural* | *Non-structural* | | |
| D1 | No | Slight | Only penetration and pollution | |
| D2 | No to slight | Moderate | Slight cracks in supporting elements Impressed doors and windows Contamination | |
| D3 | Moderate | Heavy | Major cracks and/or deformations in supporting walls and slabs Settlements | |
| D4 | Heavy | Very heavy | Structural collapse of supporting walls, slabs | |
| D5 | Very heavy | Very heavy | Collapse of the building or of major parts of the building | |



Table 5: Table of influence for indicator weighting, ranging from slight influence of an indicator (1) to extreme influence (9) (modified after Saaty (1980)).

| 1 | 2 | 3 | 4 | 5 | 6 | 7 | 8 | 9 |
|---|---|---|---|---|---|---|---|---|
| Slight influence | Slight to moderate influence | Moderate influence | Moderate to strong influence | Strong influence | Strong to very strong influence | Very strong influence | Very strong to extreme influence | Extreme influence |




**Figures**

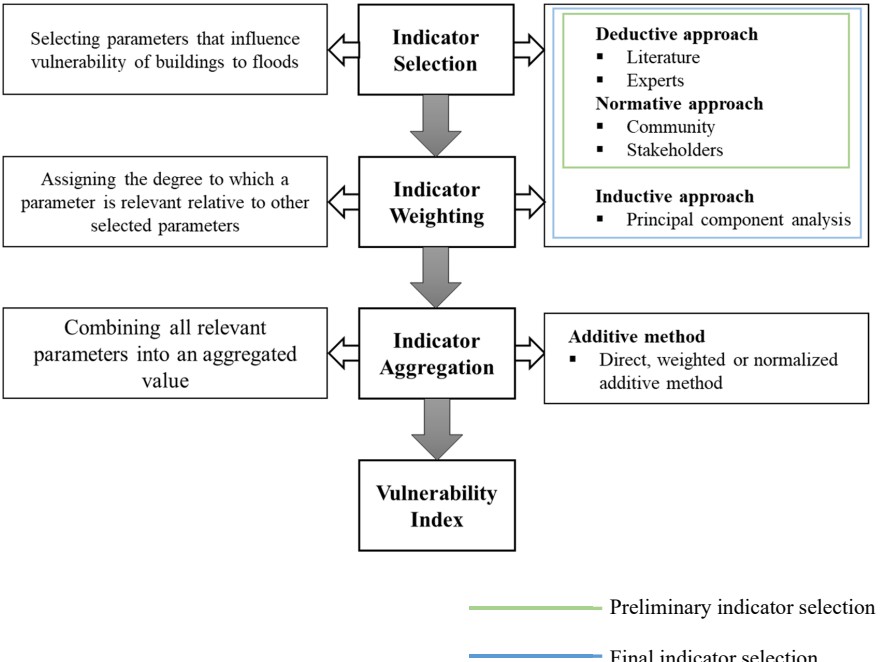

Figure 1: Steps and commonly applied methods for developing a flood vulnerability index. Steps include the indicator

selection, the indicator weighting, and the indicator aggregation. Methods are indicated for each step.



| | i | ii | iii |
|---|---|---|---|
| | | | |
| **REGION A: Aggregated Index** | 0.7 | 0.6 | 0.9 |
| Vulnerability indicators<br>▪ Building material<br>▪ Building condition<br>▪ Distance to channel<br>▪ Flood depth | Sandcrete block<br>Moderate<br>100 m<br>1 m | Sandcrete block<br>Good<br>50 m<br>1.2 m | Clay<br>Poor<br>< 20 m<br>0.60 m |
| **REGION B: Aggregated Index** | 0.5 | 0.4 | 0.7 |
| Vulnerability indicators<br>▪ Building age<br>▪ Building quality<br>▪ Sheltering effect<br>▪ Flood depth | < 10 years<br>Good<br>Complete<br>1m | 20 years<br>Moderate<br>Partial sheltering<br>1m | > 30 years<br>Poor<br>No sheltering<br>1m |
| **Maiwald and Schwarz (2015)**<br>5-category damage grade<br>Germany | **DG 2**<br>Slight cracks in supporting element | **DG 4**<br>Partial collapse of supporting element | **DG 5**<br>Collapse |
| **Ettinger et al. (2016)**<br>4-category damage grade<br>Peru | **Light**<br>Signs of impact | **Heavy**<br>Partial/Total collapse | **Heavy**<br>Partial/Total Collapse |

Figure 2: Illustration of the need for linking vulnerability index and damage grades using real damage cases (i, ii, and iii) documented after a 2017 flood in Suleja/Tafa, Nigeria, hypothetical vulnerability indicators and regions (A and B), and damage grades developed from studies by Maiwald and Schwarz (2015) and Ettinger et al. (2016).


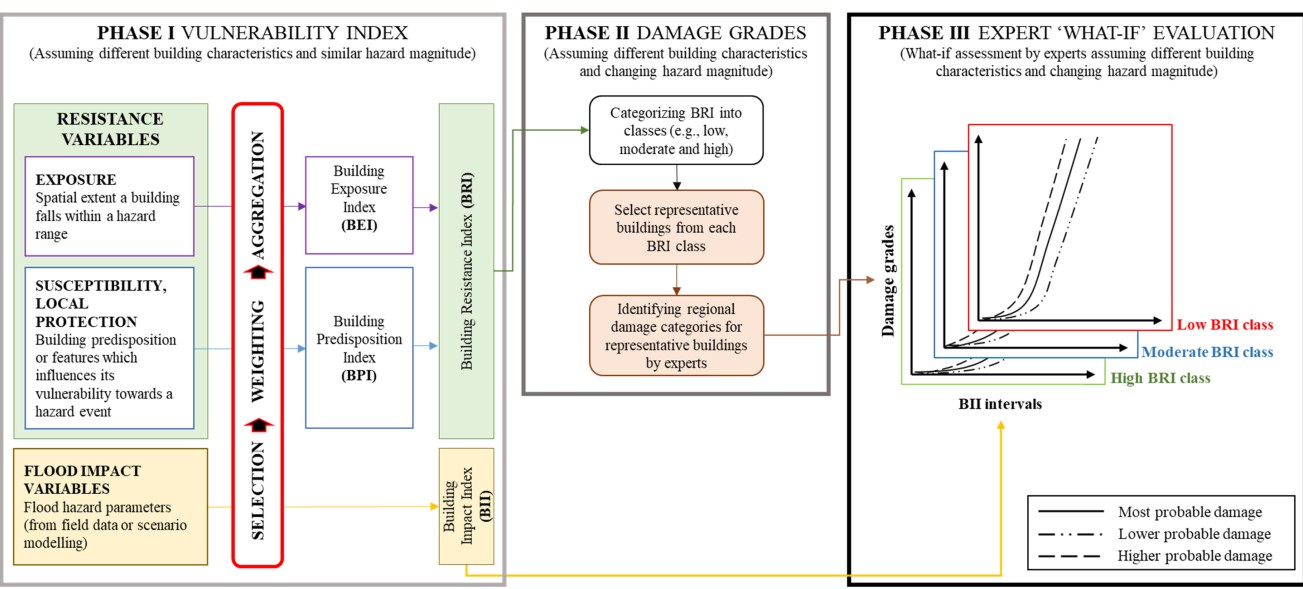


Figure 3: The proposed conceptual framework, linking vulnerability indicators to damage grades so that vulnerability to the built environment can be better assessed in data-scarce regions. The framework consists of three consecutive steps (phases) from the vulnerability index development (assuming different building characteristics but similar hazard magnitudes) to the damage grades (assuming different building characteristics and changing hazard magnitudes) and finally an expert-based "what-if"-evaluation, leading to functions linking damage grades from phase II to Building Impact Indices (BIIs) from phase I for each BRI class.