# Peer review of "A generic physical vulnerability model for floods: Review and concept for data-scarce regions"

_Natural Hazards and Earth System Sciences, 2019_

## Referee Comment (RC1) · Anonymous Referee #1 · 16 Jan 2020

The paper brings an important contribution to the field of disaster risk reduction and is worth of publication. However, an important effort of synthesis is required. Often the information is repetitive, little elaborated and some other times not relevant enough with respect to the objectives and subject of the paper. This makes difficult to review the paper. For instance, sections 2.1 and 2.2 present many subsections and secondary information which are too general and more relevant for the format of a report than for a scientific article. The authors should make an effort to reduce redundancy and secondary information to streamline the message and render the paper readable by better targeting the specific gap they are addressing.

Minor comments on the initial part of the paper: Title: if the all method is tailored only to flood perhaps include this in the title. Also, perhaps "adaptive" is little informative and generates confusion with the adaptation component frequently used in the DRR literature. I suggest using the word "generic": "A generic regional flood vulnerability assessment model: Review and concepts for data-scarce regions" Line 13: is this physical vulnerability to floods only? Perhaps add "to floods" after physical vulnerability Line 16: not clear what "local protection elements stand for in the context of that sentence Lines 61-62: insert commas after "e.g." Lines 72-78: perhaps have this paragraph in this format: "...studies earthquakes (cite cite cite), landslides (cite cite cite), tsunamis (cite cite cite) ..." and so on. Line 80: you mean "physical vulnerability assessment methods"? I'd always add "physical" to "vulnerability" to specify that you look at this type of vulnerability Line 82: "Vulnerability assessment methods are mainly used to estimate damage or loss." It's a repetition from line 68 Lines 82-83: this is a repetition from lines 35-37 Lines 82-98: perhaps connect this part on models with the previous part in which you also review methods to assess physical vulnerability. Is there any overlap? Lines 121-123: not clear Line 128: what do you mean by "combination of methods" expert based and modeling? Section 2: there is overlap and repletion with Lines 64-81. Perhaps reduce section 1 to the main points you want to bring forward in the study and move those lines to section 2. Line 162: you mean" buildings' vulnerability"? What is it meant by "framing indicator schemes"? Line 165: revise punctuation here. Line 170: use Papathoma-Koehle instead of Papathoma Line 175-177: this is a repetition from Line 165. Section 2.2.1 this section might be reduced to a sentence. There seems no need to have a separate section. Also, most of the information contained in this subsection is always consistent with the title of the section. The numbering of the section does not seem to be correct Lines 198-199: the sentence is unclear Section 2.1.4: Application of what? The title of the section is not informative enough. Overall the section seems to provide redundant information Lines 216-218: Perhaps change to "Spatial scales for assessing vulnerability can be micro-, meso- or macro". And you mean indices or indicators? Line 223: "smaller" or "bigger"?

Section 2.1.5: you use interchangeably micro, small and local. To be consistent please chose one formulation. Lines 226-227: not sure about the information provided in this sentence. . . ..

---

## Referee Comment (RC2) · Anonymous Referee #2 · 23 Jan 2020

**"general comments"**

The manuscript represents a good contribution to the understanding of natural hazards and their consequences. The presented conceptual framework aims to links vulnerability indicators with damage grades which highlights the value of damage grades in physical vulnerability assessments. A topic which is currently under-investigated. For the reader, it presents a comprehensive review on indicator-based approaches of physical vulnerability and flood damage models.

However, I have some major concerns that should be clarified and fixed before the paper can be fully accepted for publication.

First, the goals of the study are not always declared clearly. The link between the review part and the conceptual framework could be more streamlined. I suggest condensing the literature review and provide more details how to operationalize the framework including details about developed indicators.

Secondly, I am a little bit confused by using the term vulnerability which is usually broader defined and includes social, ecological and economic vulnerability. In Section 2 you mentioned a focus on physical vulnerability to floods with a specific attention to buildings. The term building vulnerability is not properly defined in the paper, and it seems that this specific element of vulnerability is a main research area. Thus, the focus of the paper needs more streamlining (title, review and framework). I doubt that the developed framework is easily transferable to social or ecological vulnerability.

Thirdly, section 5 needs more attention to explain the operationalization from concept to application. The operationalization of the framework is very conceptual, and in some aspects, it is very vague. It misses the connection to empirical indicators that builds the indices (BII, BRI, etc.) and thus shows that it can actually be applied to empirical case studies. Moreover, there are important aspects in the operationalization of damage grades that need more attention: e.g. judgement biases in the grading process, standardized training of experts and context-specific definitions of the grades etc.

Fourthly, the three phases in Section 5.2. could be better embedded in the overall picture of the paper. For example, Table 4 presents a damage grade scheme which is unclear whether the conceptual framework applies the same grading scheme or not. In Section 5.1 and Section 3.1, recommendations for best-practice are mentioned by Blong (2003b). I do not see these recommendations picked up in the framework.

Fifthly, the structure needs attention and the arguments are sometimes not placed in the right sections. In Section 2 you have too many (sub)subsection, followed by many lists with detailed arguments. Section 4 discusses the need for linking indicator and damage grades but is not clear whether it is linked to the own contributing or written as

a conclusion of the literature review. In Section 5, the author's contribution should be in the center. Explanations and smaller reviews should be avoided here. I see Section 5.1. a bit like a repetition of what is explained in section 2, 3 and 4.

Sixthly, the main contribution of the conceptual framework, which I think is the applicability in data-scare regions is not sufficiently discussed in section 2 and 3. It should be more on the point. Also, the term 'adaptability' of physical vulnerability assessments to other regions could be better picked up in the review. Are all physical vulnerability assessments adaptive regional models? Which are regional adaptive? Why? I also suggest providing more information about the specific requirements and capacity for applying them across different regions. Your tables should reflect this by focusing on these aspects.

**"specific comments"**

Paper is too long and own contribution is relatively short. Title: "adaptive" and "regional" are not well addressed in the paper. Abstract: When reading the abstract the conceptual framework is in the center, however, this is not reflected by the paper which focuses more on the literature review. Introduction: too long and broad Line 81: Unclear how social loss is defined. I do not agree that building damages are the only or one of the most important factors for social loss. In particular, if you consider that not all affected people own a building. Line 83: There is a critical difference between social and physical vulnerability assessment. You need to make clearer. Line 143: more references. Line 149: the term 'holistic' needs a proper definition. Line 256: difference between indicator and index is not defined. Line 264: it is not objectivity what you mean it is comprehensiveness. Objectivity is needed for every selected indicator. Line 222ff. Indicator weighting: statistical weighting based on data can explain the consistency and inference of indicators but cannot be used for an appropriate weighting of importance or measurability of the indicators. This should be mentioned at the beginning. Line 571: unclear if this is part of the framework or part of literature review. Line 701: Do you applied field surveys or remote sensing? When discussion about different option should be mentioned in the review section. Conclusion: Be more precise about the key messages both from the review and from the conceptual framework. An outlook of how the framework will be applied is also helpful for presenting the relevance. Please elaborate on the link between the relevance for risk reduction methods in developing countries and the data scarcity and barriers in collecting data in most of the developing countries?

**"technical corrections"**

Line 249: this sentence needs a reference Line 322: acronyms are not defined in table figure caption. Line 329f. Sentence is unclear also the example does not seem to make sense. Line 443: reference missing Linen 415f: unclear sentence Line 650, 661 and 680: fourth level of headline should be avoided.

---

## Author Comment (AC1) · 4 Mar 2020

The authors would like to thank Referee #1 for his/her insightful comments. Below, answers to the concerns raised are provided step-by-step.

The paper brings an important contribution to the field of disaster risk reduction and is worth of publication. However, an important effort of synthesis is required. Often the information is repetitive, little elaborated and some other times not relevant enough with respect to the objectives and subject of the paper. This makes difficult to review the paper. For instance, sections 2.1 and 2.2 present many subsections and secondary information which are too general and more relevant for the format of a report than

[Figure]

for a scientific article. The authors should make an effort to reduce redundancy and secondary information to streamline the message and render the paper readable by better targeting the specific gap they are addressing.

Authors: Thank you for the comment. We will make a substantial effort to streamline, reorganize and reduce the manuscript (especially in sections 1, 2 and 3). Suggested changes will include the following; Section 1 - Repetitive information will be removed and details provided will be limited to a general overview and the gaps we plan to address Section 2 - Subsections in 2.1 will be contracted into one paragraph (background). Section 2.2 (application) will be reduced to better focus the methodology on expert-based approaches which will be implemented in the conceptual framework. Section 2.3 will focus on current challenges/gaps and specific areas the conceptual framework will address Section 3 – Information provided will be reduced and reorganized into the background (section 3.1), application (section 3.2) and challenges/gaps and specific areas the conceptual framework seeks to address Section 4, 5, 6 – Information will be streamlined and reorganized.

Title: if the all method is tailored only to flood perhaps include this in the title. Also, perhaps "adaptive" is little informative and generates confusion with the adaptation component frequently used in the DRR literature. I suggest using the word "generic": "A generic regional flood vulnerability assessment model: Review and concepts for data-scarce regions"

Authors: Thank you for the suggestion. We will change the title to "A generic physical vulnerability model for floods: review and concepts for data-scarce regions"

Line 13: is this physical vulnerability to floods only? Perhaps add "to floods" after physical vulnerability

Authors: Changes will be made accordingly. See also a suggestion for a new title.

Line 16: not clear what "local protection elements stand for in the context of that sentence.

Authors: Local protection in the context of our study was defined (now slightly modified) in Line 508-513 as deliberate or non-deliberate measures that are put in place and can reduce the impact of floods on a building. These measures can be directly included in the building structure e.g. elevation of the entrance door, or measures located in the immediate surrounding of a building. While many local structural protection measures may not be primarily constructed as a protection mechanism against floods, they reduce the impact of floods on a building e.g. fencing wall (Attems et al., 2020; Holub et al., 2012; Holub and Fuchs, 2008). Due to a suggestion by referee #2, the sentence will be removed from the abstract.

Lines 61-62: insert commas after "e.g."

Authors: Thank you for the comment. Changes will be made accordingly.

Lines 72-78: perhaps have this paragraph in this format: "...studies earthquakes (cite cite cite), landslides (cite cite cite), tsunamis (cite cite cite)..." and so on.

Authors: Thank you for the comment. Changes will be made accordingly.

Line 80: you mean "physical vulnerability assessment methods"? I'd always add "physical" to "vulnerability" to specify that you look at this type of vulnerability

Authors: Thank you for the comment. We will add the term 'physical' vulnerability to other parts of the manuscript to be more specific.

Line 82: "Vulnerability assessment methods are mainly used to estimate damage or loss." It's a repetition from line 68

Authors: Thank you for the comment. Changes will be made accordingly.

Lines 82-83: this is a repetition from lines 35-37

Authors: Thank you for the comment. Changes will be made accordingly.

Lines 82-98: perhaps connect this part on models with the previous part in which you also review methods to assess physical vulnerability. Is there any overlap?

Authors: Thank you for the comment. Since we were planning on combining the vulnerability indicator method and damage grades, the idea was to introduce them separately. Firstly, we review 'approaches' for physical vulnerability assessment (vulnerability curves, indicators, matrices, and multivariate methods) and secondly we report on their application (monetary loss and damage grade prediction). However, we will combine the parts on commonly applied physical vulnerability assessment methods which are also used for damage grade prediction (vulnerability curves and the multivariate method) and then discuss the vulnerability indicator method separately. The suggested change in linking the paragraph: "Generally, both the stage-damage curves and the multivariate methods have been used to predict flood damage. This ability to predict damage is increasingly seen as an important step towards disaster risk reduction (Merz et al., 2010). These models used to predict building damage due to flood impact are commonly referred to as flood damage models."

Lines 121-123: not clear

Authors: We refer to the uncertainties resulting from two factors, (1) the use of vulnerability curves from other regions which do not have comparable building or hazard characteristics and (2) from the use of meso-scale aggregated data which can overlook certain characteristics of a community that is only assessable through micro-scale assessment. Thus, lines 121-123 highlights that these two factors can contribute to higher uncertainties. We will streamline this in a revised version.

Line 128: what do you mean by "combination of methods" expert based and modeling?

Authors: By a combination of methods, we mean merging (or integrating) approaches or techniques from two different physical vulnerability assessment methods into a new method. For example, combining vulnerability curves (data-driven) with vulnerability indicators (expert-based) as demonstrated in (Godfrey et al., 2015). We will clarify this

in a revised version of the manuscript.

Section 2: there is overlap and repletion with Lines 64-81. Perhaps reduce section 1 to the main points you want to bring forward in the study and move those lines to section 2.

Authors: Thank you for the comment. Section 1 will be reduced and further overview of vulnerability indicators will be moved to section 2.

Line 162: you mean" buildings' vulnerability"?

Authors: Thank you for the comment. We meant the vulnerability of buildings and we will undertake necessary changes in a revised version of the manuscript.

What is it meant by "framing indicator schemes"?

Authors: Framing indicator schemes here means 'setting the underlying (theoretical) framework for indicators'. We will clarify this in a revised version.

Line 165: revise punctuation here.

Authors: Thank you for the comment. Changes will be made accordingly.

Line 170: use Papathoma-Koehle instead of Papathoma

Authors: Thank you for the comment. Changes will be made accordingly.

Line 175-177: this is a repetition from Line 165.

Authors: Thank you for the comment. The repeated part will be deleted in a revised manuscript version.

Section 2.2.1 this section might be reduced to a sentence. There seems no need to have a separate section. Also, most of the information contained in this subsection is always consistent with the title of the section. The numbering of the section does not seem to be correct

[Figure]

Authors: Thank you for the comment. In section 2.2.1, it was important to highlight three main issues relating to indicator selection; (a) Recommended criteria for selecting indicators (b) number of indicators, and (c) different approaches and stages used for selecting indicators. The conceptual framework to be introduced in section 5 will require such information as a basis for future studies that implement the framework. We will streamline and reduce the details provided in the section.

Lines 198-199: the sentence is unclear

Authors: According to Birkmann (2006), the choice of including different dimensions of vulnerability might be related to data availability. For example, in countries where there are regularly-updated and available demographic data (e.g., income level, gender, age, employment, etc.) it is common to find studies that combine physical and social vulnerability. We will clarify this in a revised version.

Section 2.1.4: Application of what? The title of the section is not informative enough. Overall the section seems to provide redundant information

Authors: Thank you for your comment. We meant 'Application of the vulnerability index in the risk cycle'. Section 2.1.4 provides information that the vulnerability index can be applied to different stages of the risk cycle. For example, it can be applied for disaster preparedness, disaster response, and disaster mitigation. In addition, it highlights that most studies use developed indices for preparedness and mitigation. Generally, knowledge of the application of the index will guide the selection and weighting of the indicators. We will streamline and reduce the information provided in the section.

Lines 216-218: Perhaps change to "Spatial scales for assessing vulnerability can be micro-, meso- or macro".

Authors: Thank you for the comment. Changes will be made accordingly.

And you mean indices or indicators?

Authors: Thank you for the comment. "Indicators" is the correct term. Changes will be

made accordingly.

Line 223: "smaller" or "bigger"?

Authors: Thank you for the comment. The correct term is 'bigger'. Changes will be made accordingly.

Section 2.1.5: you use interchangeably micro, small and local. To be consistent please chose one formulation.

Authors: Thank you for the comment. Changes will be made accordingly.

Lines 226-227: not sure about the information provided in this sentence.

Authors: According to Eriksen and Kelly (2007), the basic scale of vulnerability is the local scale since it is at this scale that communities differ. Consequently, assessing vulnerability at either a regional or national scale leads to information loss from averaging or aggregation. Due to this loss of information, vulnerability assessment at a higher scale (macro or meso) requires careful interpretation. We will modify the sentence to clarify the intended idea. The suggested change is "Since the basic scale of vulnerability, at which communities differ, is at the micro-scale (Eriksen and Kelly, 2007), care must be taken when aggregating information for meso- and macro-scale assessment". Changes will be made accordingly.

References

Attems, M., Thaler, T., Genovese, E. and Fuchs, S.: Implementation of property‐level flood risk adaptation (PLFRA) measures: Choices and decisions, Wiley Interdiscip. Rev. Water, 7(1), e1404, 2020. Eriksen, S. H. and Kelly, P. M.: Developing credible vulnerability indicators for climate adaptation policy assessment, Mitig. Adapt. Strateg. Glob. Chang., 12(4), 495–524, doi:10.1007/s11027-006-3460-6, 2007. Godfrey, A., Ciurean, R. L., van Westen, C. J., Kingma, N. C. and Glade, T.: Assessing vulnerability of buildings to hydro-meteorological hazards using an expert based approach - An application in Nehoiu Valley, Romania, Int. J. Disaster Risk Reduct., 13,

229–241, doi:10.1016/j.ijdrr.2015.06.001, 2015. Holub, M. and Fuchs, S.: Benefits of local structural protection to mitigate torrent-related hazards, WIT Trans. Inf. Commun. Technol., 39, 401–411, 2008. Holub, M., Suda, J. and Fuchs, S.: Mountain hazards: reducing vulnerability by adapted building design, Environ. Earth Sci., 66(7), 1853–1870, 2012.

---

## Author Comment (AC2) · 4 Mar 2020

The authors would like to thank Referee #2 for his/her insightful comments. Below, answers to the concerns raised are provided step-by-step.

"general comments"

The manuscript represents a good contribution to the understanding of natural hazards and their consequences. The presented conceptual framework aims to links vulnerability indicators with damage grades which highlights the value of damage grades in physical vulnerability assessments. A topic which is currently under-investigated. For

the reader, it presents a comprehensive review on indicator-based approaches of physical vulnerability and flood damage models. However, I have some major concerns that should be clarified and fixed before the paper can be fully accepted for publication. First, the goals of the study are not always declared clearly. The link between the review part and the conceptual framework could be more streamlined. I suggest condensing the literature review and provide more details how to operationalize the framework including details about developed indicators.

Authors: Thank you for your comment. The manuscript will be streamlined to reduce the review and provide more details on the conceptual framework. Sections 1, 2, and 3 will be streamlined and reduced, and we will take up the highlighted issues in sections 4 and 5 and provided a more detailed discussion.

Secondly, I am a little bit confused by using the term vulnerability which is usually broader defined and includes social, ecological and economic vulnerability. In Section 2 you mentioned a focus on physical vulnerability to floods with a specific attention to buildings. The term building vulnerability is not properly defined in the paper, and it seems that this specific element of vulnerability is a main research area. Thus, the focus of the paper needs more streamlining (title, review and framework). I doubt that the developed framework is easily transferable to social or ecological vulnerability.

Authors: Thank you for your comment. The manuscript will be streamlined to better adapt the term physical (building) vulnerability in the title and review part. The suggested title now is "A generic physical vulnerability model for floods: Review and concepts for data-scarce regions". The term vulnerability was not defined in the earlier sections (section 2, 3) because the reviewed studies used different definitions of vulnerability. However, the UNISDR (2009) definition was given in line 30. For our framework, we adopted a specific definition for vulnerability as stated in section 5.1. Nevertheless, we will streamline a revised version of the manuscript so that it becomes more accessible.

Thirdly, section 5 needs more attention to explain the operationalization from concept to application. The operationalization of the framework is very conceptual, and in some aspects, it is very vague. It misses the connection to empirical indicators that builds the indices (BII, BRI, etc.) and thus shows that it can actually be applied to empirical case studies. Moreover, there are important aspects in the operationalization of damage grades that need more attention: e.g. judgement biases in the grading process, standardized training of experts and context-specific definitions of the grades etc.

Authors: The proposed indices (BII, BRI ) are aggregations of the selected and weighted indicators. As shown in Figure 3 (Phase 3), the BRI and BII form the basis for the synthetic curve used for predicting damage grades. The developed damage grades are meant to offer simplistic comparisons and not precise predictions. Since it is an expert-based approach, judgment biases cannot be completely eliminated. However, in order for the proposed method to capture the actual damage range, we propose the use of the three damage states (most probable, lower probable and higher probable damage). Furthermore, as described in section 5.2.2, based on a recommendation by Grünthal et al. (1998) and Maiwald and Schwarz (2015), the definition of damage grades is not only based on damage pattern but on proportion. This ensures that damage grades are representative of the actual distribution of damage patterns in the region. We will make these recommendations clearer in the manuscript. We will also highlight that damage grades are regionally adaptive. That is, they are based on commonly-observed features within a region, hence, context-specific.

Fourthly, the three phases in Section 5.2. could be better embedded in the overall picture of the paper. For example, Table 4 presents a damage grade scheme which is unclear whether the conceptual framework applies the same grading scheme or not. In Section 5.1 and Section 3.1, recommendations for best-practice are mentioned by Blong (2003b). I do not see these recommendations picked up in the framework.

Authors: Thank you for the comment. We will adapt sections 2 and 3 to reflect the methods used in section 5.2. The suggested change will have the same format for

both indicators (section 2) and damage grades (section 3), and present a background, application, and challenges for data-scarce regions. Further, we plan to extend discussions on identified challenges in section 4 and a proposed idea for the new framework in section 5. Table 4 serves to present an example of damage grades developed for Germany by Maiwald and Schwarz (2007). The conceptual framework recommends that damage grades are developed in a regional context. Frequently observed damage patterns within the region are to be used for developing the damage grades. In Section 5.2.2 we outlined that the main aim of this step is to identify commonly-observed damage patterns within a region and categorize them into classes. The damage grades developed by Maiwald and Schwarz (2007) which our study was based upon were developed using the recommendations by Grünthal (1993). Consequently, techniques described in section 5.2.2 systematically integrate the recommendations. We will clarify this in a revised version of the manuscript.

Fifthly, the structure needs attention and the arguments are sometimes not placed in the right sections. In Section 2 you have too many (sub)subsection, followed by many lists with detailed arguments. Section 4 discusses the need for linking indicator and damage grades but is not clear whether it is linked to the own contributing or written as a conclusion of the literature review. In Section 5, the author's contribution should be in the center. Explanations and smaller reviews should be avoided here. I see Section 5.1. a bit like a repetition of what is explained in section 2, 3 and 4.

Authors: Thank you for the comment. Sub(sections) and details in section 2 will be reduced. Section 4 is not a direct conclusion of the reviews presented in sections 2 and 3. Rather, section 4, draws from challenges highlighted in the individual methods and illustrates the added value for combining them in data-scarce regions. This illustration was carried out by using three observed damage cases from Nigeria during the 2017 flood. For section 5, we will reduce the information to better focus the content on our contribution. Section 5.1 was not a repetition but background information which includes definitions we adopt for our study (e.g., vulnerability, exposure, susceptibil-

ity, local protection). This information was necessary before introducing section 5.2 "Operationalizing the framework". The information was not provided in sections 2 and 3 (since they are reviews from different studies) or in section 4 (since it focuses on the linkage of vulnerability indicators and damage grades). We will make necessary adjustments during revision, see also comments to referee #1.

Sixthly, the main contribution of the conceptual framework, which I think is the applicability in data-scare regions is not sufficiently discussed in section 2 and 3. It should be more on the point. Also, the term 'adaptability' of physical vulnerability assessments to other regions could be better picked up in the review. Are all physical vulnerability assessments adaptive regional models? Which are regional adaptive? Why? I also suggest providing more information about the specific requirements and capacity for applying them across different regions. Your tables should reflect this by focusing on these aspects.

Authors: Thank you for your comment. We will streamline the contents of sections 2 and 3 to better address applications of each method for data-scarce regions. The suggested modification includes introducing challenges (for data-scarce regions) in both sections 2.3 and 3.3. These will serve as a background for research gaps that will be taken up and addressed in the framework. Another suggested change is to focus the review in section 2 on the deductive and normative method since they are more suitable for data-scarce areas. Based on recommendations from referee #1, we will change the term 'adaptable' to 'generic' in the title. This was to avoid confusion with the adaption component of the Disaster Risk Reduction (DRR) literature. Nonetheless, we will address the need for making the indicators and damage grades context-based.

"specific comments"

Paper is too long and own contribution is relatively short.

Authors: Thank you for your comment. We will make substantial effort to streamline sections 1-5 and reduce the content of the manuscript. We will also better highlight our

contribution in the revised manuscript.

Title: "adaptive" and "regional" are not well addressed in the paper.

Authors: Based on a recommendation from Referee #1, we will replace the term 'adaptive' with the word 'generic' in order to avoid confusion with the adaptation component frequently used in the DRR literature. We will better address adaptability to the regional situation a revised version of the manuscript.

Abstract: When reading the abstract the conceptual framework is in the center, however, this is not reflected by the paper which focuses more on the literature review.

Authors: The abstract will be reformulated to balance both the review and conceptual framework. The suggested new abstract is "The use of different physical vulnerability assessment methods have evolved over the years from the traditional single-parameter stage-damage curves to multi-parameter approaches like the multivariate and vulnerability indicator methods. However, despite wide applications of these methods in assessing future flood risk, a gap remains in their application to data-scarce regions. Assessing physical vulnerability against future risks is even more critical for data-scarce regions, which are mostly are mostly areas with limited capacity to cope with disasters. To address this gap, we propose an expert based framework to link vulnerability indicators (integrating major damage drivers) with damage grades (integrating frequently observed damage patterns). To do this, we review current studies on physical vulnerability to floods indicators and flood damage models to gain insights on best practices. Thereafter, we propose a new conceptual framework to address selected gaps in literature. The conceptual framework is operationalized using three phases (i) developing the vulnerability index, (ii) identifying regional damage grades, and (iii) linking developed index classes with damage patterns utilizing the synthetic what-if analysis. The new framework constitutes a basic first step for enhancing damage prediction to support risk reduction in data-scarce regions. The framework is adaptable to different data-scarce environments and can integrate future changes in damage drivers and

damage grades".

Introduction: too long and broad

Authors: Thank you for your comment. We will streamline and reduce the details provided in the introduction.

Line 81: Unclear how social loss is defined. I do not agree that building damages are the only or one of the most important factors for social loss. In particular, if you consider that not all affected people own a building.

Authors: Thank you for your comment. We will provide a definition of social loss and better address this concern.

Line 83: There is a critical difference between social and physical vulnerability assessment. You need to make clearer.

Authors: Thank you for your comment. In a revised version, we will better address this concern and more strongly focus on physical vulnerability.

Line 143: more references.

Authors: Thank you for your comment. Changes will be made accordingly.

Line 149: the term 'holistic' needs a proper definition.

Authors: Thank you for the comment. We will include the term 'comprehensive' to denote an assessment that considers all possible influencing parameters.

Line 256: difference between indicator and index is not defined.

Authors: Thank you for your comment. Suggested modification in the sentence reads "Generally, the aim of indicators is to simplify a concept through the use of an index (Heink and Kowarik, 2010; Hinkel, 2011). A vulnerability index is obtained by selecting, weighting and aggregating vulnerability indicators."

Line 264: it is not objectivity what you mean it is comprehensiveness. Objectivity is

needed for every selected indicator.

Authors: Thank you for your comment. Changes will be made accordingly.

Line 222ff. Indicator weighting: statistical weighting based on data can explain the consistency and inference of indicators but cannot be used for an appropriate weighting of importance or measurability of the indicators. This should be mentioned at the beginning.

Authors: Comment is not very clear. We will add an explanation of the indicator weighting in a revised version of the manuscript.

Line 571: unclear if this is part of the framework or part of literature review.

Authors: Section 4 focuses mainly on the added value for linking vulnerability indicators and damage grades for data-scarce regions. It is not meant to be part of the review or the new conceptual framework, rather, it is the authors' contribution to illustrate the added value of combining two physical vulnerability assessment methods. This was demonstrated by using a hypothetically developed vulnerability index for two regions in combination with three building damage data from Nigeria. This section uses practical examples to bridge the gap between identified challenges/gaps presented in the reviews (in sections 2 and 3) and conceptual framework (section 5).

Line 701: Do you applied field surveys or remote sensing? When discussion about different option should be mentioned in the review section.

Authors: Carrying out a field survey to determine building typology and characteristics will be the most preferred option for such data collection because experts can have a first-hand impression of the local situation and can identify, in detail, construction features or qualities, which will determine how building representatives are selected. However, due to time and financial constraints, carrying out a field survey is not always feasible. Hence, the use of remote sensing is encouraged as an option, especially in a meso-(or macro-) scale study region,. Consequently, the authors recommended the

study by Blanco-Vogt and Schanze (2014) which focus on extracting building representatives for meso-(or macro-) scale assessment.

Conclusion: Be more precise about the key messages both from the review and from the conceptual framework. An outlook of how the framework will be applied is also helpful for presenting the relevance. Please elaborate on the link between the relevance for risk reduction methods in developing countries and the data scarcity and barriers in collecting data in most of the developing countries?

Authors: Thank you for your comment. We will elaborate on the relevance of this linkage and current challenges in data-scarce regions (see also suggestions of referee #1).

"technical corrections"

Line 249: this sentence needs a reference

Authors: Thank you for your comment. Changes will be made accordingly.

Line 322: acronyms are not defined in table figure caption.

Authors: Thank you for your comment. Changes will be made accordingly

Line 329f. Sentence is unclear also the example does not seem to make sense.

Authors: Thank you for your comment. The example will be rephrased to better capture the intended purpose. The suggested change is: "For example, if we assume the same hazard level impacting a reinforced concrete and clay building, it is most likely that the clay building will incur higher damage than the reinforced concrete building. Therefore, experts my score a reinforced concrete building as less vulnerable than the clay building. However, in order to assign a value that qualifies the extent to which the reinforced concrete building is less vulnerable than the clay building (e.g., moderate, high, very high, etc.), expert knowledge will be required. This is because such assessment will require not only damage data but other factors such as common quality of construction

types and material".

Line 443: reference missing

Authors: Thank you for your comment. Changes will be made accordingly.

Linen 415f:

Authors: Unclear comment.

Line 650, 661 and 680: fourth level of headline should be avoided.

Authors: Thank you for your comment. Changes will be made accordingly.

---

## Author Response (AR1)

**RESPONSE TO COMMENTS FROM REFEREE #1**

Authors: We would like to thank Referee #1 for his/her insightful comments. Below, answers to the concerns raised are provided point-by-point.

The paper brings an important contribution to the field of disaster risk reduction and is worth of publication. However, an important effort of synthesis is required. Often the information is repetitive, little elaborated and some other times not relevant enough with respect to the objectives and subject of the paper. This makes difficult to review the paper. For instance, sections 2.1 and 2.2 present many subsections and secondary information which are too general and more relevant for the format of a report than for a scientific article. The authors should make an effort to reduce redundancy and secondary information to streamline the message and render the paper readable by better targeting the specific gap they are addressing.

Authors: Thank you for the comment. Substantial effort has been made to streamline, reorganize and reduce the manuscript (especially in sections 1, 2 and 3). Suggested changes included the following; Section 1 – information is streamlined and reorganized and details provided are now limited to a general overview and the gaps we plan to address, Section 2 – Table 1 has been moved to the appendix, subsections in 2.1.1 – 2.1.6 have been contracted into one paragraph (background). Section 2.2 (application) is reduced to better focus the methodology on expert-based approaches which will be implemented in the conceptual framework, Section 2.3 focuses on current challenges/gaps and specific areas the conceptual framework seeks to address, Section 3 – Information provided is reduced and reorganized into the background (section 3.1), application (section 3.2), and challenges/gaps the conceptual framework seeks to address. Sections 4, 5, 6 – Information is streamlined and reorganized.

Referee #1: Title: if the all method is tailored only to flood perhaps include this in the title. Also, perhaps "adaptive" is little informative and generates confusion with the adaptation component frequently used in the DRR literature. I suggest using the word "generic": "A generic regional flood vulnerability assessment model: Review and concepts for data-scarce regions"

Authors: Thank you for the suggestion. The title has been modified (also in consideration of comments from reviewer #2). The new title is "A generic physical vulnerability model for floods: Review and concept for data-scarce regions"

Line 13: is this physical vulnerability to floods only? Perhaps add "to floods" after physical vulnerability

Authors: Thank you for your comment. Changes have been made accordingly.

Line 16: not clear what "local protection elements stand for in the context of that sentence.

Authors: Local protection in the context of our study was defined (and now slightly modified) in lines 480-485 as "deliberate or non-deliberate measures that are put in place and can reduce the impact of floods on a building. These measures can be directly included in the building structure e.g. elevation of the entrance door, or measures located in the immediate surrounding of a building. While many local structural protection measures may not be primarily constructed as a protection mechanism against floods, they reduce the impact of floods on a building e.g. fencing wall (Attems et al., 2020; Holub and Fuchs, 2008)". Due to a suggestion by referee #2, the sentence has been removed from the abstract.

Lines 61-62: insert commas after "e.g."

Authors: Thank you for the comment. Changes have been made accordingly.

Lines 72-78: perhaps have this paragraph in this format: "…studies earthquakes (cite cite cite), landslides (cite cite cite), tsunamis (cite cite cite)…" and so on.

Authors: Thank you for the comment. Changes have been made accordingly. The sentence was later removed due to the necessary reorganization and streamlining of the manuscript.

Line 80: you mean "physical vulnerability assessment methods"? I'd always add "physical" to "vulnerability" to specify that you look at this type of vulnerability

Authors: Thank you for the comment. The term 'physical' is added to vulnerability (also in other parts of the manuscript) to be more specific.

Line 82: "Vulnerability assessment methods are mainly used to estimate damage or loss." It's a repetition from line 68

Authors: Thank you for the comment. Changes have been made accordingly.

Lines 82-83: this is a repetition from lines 35-37

Authors: Thank you for the comment. Changes have been made accordingly.

Lines 82-98: perhaps connect this part on models with the previous part in which you also review methods to assess physical vulnerability. Is there any overlap?

Authors: Thank you for the comment. Since the paper focuses on combining the vulnerability indicator method and damage grades, the idea was to introduce them separately. Consequently, we highlight approaches for physical vulnerability assessment (stage damage curves, indicators, matrices, and multivariate methods) at first, and secondly, we report on their application (e.g., monetary loss and damage grade prediction). However, we now combine the parts on commonly applied physical vulnerability assessment methods which are also used for damage grade prediction (stage-damage curves and the multivariate method) and then discuss the vulnerability indicator method separately.

The suggested change in linking the paragraphs (currently in lines 54-56) reads: "Generally, both stage-damage curves and multivariate methods are used to predict flood damage. This ability to predict damage is increasingly seen as an important step towards disaster risk reduction (Merz et al., 2010). Stage-damage curves and multivariate methods used for damage prediction are commonly referred to as flood damage models".

Lines 121-123: not clear

Authors: We refer to the uncertainties resulting from two factors, (1) the use of stage-damage curves from regions which do not have comparable building or hazard characteristics and (2) from the use of meso-scale aggregated data which can overlook certain characteristics of a community that is only assessable through micro-scale assessment. Thus, lines 121-123 highlights that these two factors can contribute to higher uncertainties. We have streamlined this part in the revised version of the manuscript.

Line 128: what do you mean by "combination of methods" expert based and modeling?

Authors: By a combination of methods, we mean merging (or integrating) approaches or techniques from two different physical vulnerability assessment methods into a new method. For example, combining stage-damage curves (data-driven) with vulnerability indicators (expert-based) as demonstrated in Godfrey et al. (2015). A slightly modified version of the sentence, now in lines 82 – 86, reads "Papathoma-Köhle et al. (2017) recommended a combination of physical vulnerability assessment methods to take advantage of their individual strengths while minimizing their weaknesses. A combination of methods here refers to the integration of approaches (or techniques) from two different physical vulnerability assessment methods into one method (or model). Such a combination of methods that utilize expert-based approaches in place of data-driven methods might provide a desirable compromise for data-scarce regions."

Section 2: there is overlap and repletion with Lines 64-81. Perhaps reduce section 1 to the main points you want to bring forward in the study and move those lines to section 2.

Authors: Thank you for the comment. Section 1 has been reduced and details on vulnerability indicators have been moved to section 2.

Line 162: you mean" buildings' vulnerability"?

Authors: Thank you for the comment. We meant the vulnerability of buildings. Necessary changes have been undertaken in the revised version of the manuscript in line 114.

What is it meant by "framing indicator schemes"?

Authors: Framing indicator schemes here means setting the underlying (theoretical) framework for indicators. The phrase will be clarified in the revised version.

Line 165: revise punctuation here.

Authors: Thank you for the comment. Changes have been made accordingly.

Line 170: use Papathoma-Koehle instead of Papathoma

Authors: Thank you for the comment. Changes have been made accordingly.

Line 175-177: this is a repetition from Line 165.

Authors: Thank you for the comment. The repeated part has been deleted in the revised version of the manuscript.

Section 2.2.1 this section might be reduced to a sentence. There seems no need to have a separate section. Also, most of the information contained in this subsection is always consistent with the title of the section. The numbering of the section does not seem to be correct

Authors: Thank you for the comment. In section 2.2.1, it was important to highlight three main issues relating to indicator selection; (1) Recommended criteria for selecting indicators (2) number of indicators, and (3) different approaches and stages used for selecting indicators. The conceptual framework to be introduced in section 5 will require such information as a basis for future studies that implement the framework. Section 2 has been streamlined and details provided have been reduced to focus on deductive methods.

Lines 198-199: the sentence is unclear

Authors: According to Birkmann (2006), the choice of including different dimensions of vulnerability might be related to data availability. For example, in countries where there are regularly-updated and available demographic data (e.g., income level,

gender, age, employment, etc.), it is common to find studies that combine physical and social vulnerability. This sentence has been removed due to the necessary reorganization and streamlining of the manuscript.

Section 2.1.4: Application of what? The title of the section is not informative enough. Overall the section seems to provide redundant information
Authors: Thank you for your comment. We meant 'Application of the vulnerability index in the risk cycle'. Section 2.1.4 provides information that the vulnerability index can be applied to different stages of the risk cycle. For example, it can be applied for disaster preparedness, disaster response, and disaster mitigation. In addition, it highlights that most studies use developed indices for preparedness and mitigation. Generally, knowledge of the application of the index will guide the selection and weighting of the indicators. Section 2.1.4 has been removed due to the necessary reorganization and streamlining of the manuscript.

Lines 216-218: Perhaps change to "Spatial scales for assessing vulnerability can be micro-, meso- or macro".
Authors: Thank you for the comment. Changes have been made accordingly.

And you mean indices or indicators?
Authors: Thank you for the comment. "Indicators" is the correct term. Changes have been made accordingly.

Line 223: "smaller" or "bigger"?
Authors: Thank you for the comment. The correct term is 'bigger'. Changes have been made accordingly.

Section 2.1.5: you use interchangeably micro, small and local. To be consistent please chose one formulation.
Authors: Thank you for the comment. Changes have been made accordingly.

Lines 226-227: not sure about the information provided in this sentence.
Authors: According to Eriksen and Kelly (2007), the basic scale of vulnerability is the micro-(local-) scale since it is at this scale that communities differ (e.g., difference in building material, quality or local protection measures). Consequently, assessing vulnerability at either a regional or national scale leads to information loss due to averaging or aggregation. Consequently, vulnerability assessment at a bigger scale (macro- or meso-) requires careful interpretation. This sentence has been removed due to the necessary reorganization and streamlining of the manuscript.

**RESPONSE TO COMMENTS FROM REFEREE #2**

Authors: We would like to thank Referee #2 for his/her insightful comments. Below, answers to the concerns raised are provided step-by-step.

**"general comments"**

The manuscript represents a good contribution to the understanding of natural hazards and their consequences. The presented conceptual framework aims to links vulnerability indicators with damage grades which highlights the value of damage grades in physical vulnerability assessments. A topic which is currently under-investigated. For the reader, it presents a comprehensive review on indicator-based approaches of physical vulnerability and flood damage models. However, I have some major concerns that should be clarified and fixed before the paper can be fully accepted for publication. First, the goals of the study are not always declared clearly. The link between the review part and the conceptual framework could be more streamlined. I suggest condensing the literature review and provide more details how to operationalize the framework including details about developed indicators.
Authors: Thank you for your comment. The manuscript has been streamlined to reduce the review and provide more details on the conceptual framework. Sections 1, 2, and 3 have been streamlined and reduced, and we have taken up the highlighted issues in sections 4 and 5 and provided a more detailed discussion.

Secondly, I am a little bit confused by using the term vulnerability which is usually broader defined and includes social, ecological and economic vulnerability. In Section 2 you mentioned a focus on physical vulnerability to floods with a specific attention to buildings. The term building vulnerability is not properly defined in the paper, and it seems that this specific element of vulnerability is a main research area. Thus, the focus of the paper needs more streamlining (title, review and framework). I doubt that the developed framework is easily transferable to social or ecological vulnerability.
Authors: Thank you for your comment. The manuscript has been streamlined to better adapt the term physical (building) vulnerability in the title and review part. The suggested title now is "A generic physical vulnerability model for floods: Review

and concepts for data-scarce regions". The term building vulnerability was not defined in the earlier sections (section 2, 3) because the reviewed studies used different definitions of vulnerability. A general definition was given for vulnerability in section 1 as the degree of resistance to impacts (line 2 – 3) "Communities with a low resistance to impacts of hazards are often referred to as vulnerable". For our framework, we mention that the focus is on physical vulnerability for buildings. We adopted a specific definition for (building) vulnerability as stated in section 5.1 (lines 466 - 467) "Vulnerability: The degree to which an exposed building will experience damage from flood hazards under certain conditions of exposure, susceptibility and resilience (adapted after Balica et al., 2009)". We have streamlined a revised version of the manuscript so that it becomes clearer.

Thirdly, section 5 needs more attention to explain the operationalization from concept to application. The operationalization of the framework is very conceptual, and in some aspects, it is very vague. It misses the connection to empirical indicators that builds the indices (BII, BRI, etc.) and thus shows that it can actually be applied to empirical case studies. Moreover, there are important aspects in the operationalization of damage grades that need more attention: e.g. judgement biases in the grading process, standardized training of experts and context-specific definitions of the grades etc.

Authors: The proposed indices (BII, BRI ) are aggregations of the selected and weighted indicators. Also, as shown in Figure 3 (Phase 3), the BRI and BII form the basis for the proposed synthetic curve(s) used for predicting damage grades. The developed damage grades are meant to offer simplistic comparisons and not precise predictions. Since it is an expert-based approach, judgment biases cannot be completely eliminated. However, in order for the proposed method to capture the actual damage range, we propose the use of the three damage states (most probable, lower probable and higher probable damage). Furthermore, as described in section 5.2.2, based on a recommendation by Grünthel et al. (1998) and Maiwald and Schwarz (2015), the definition of damage grades is not only based on damage patterns but on proportion. This ensures that damage grades are representative of the actual distribution of damage patterns in the region. Since within the scope of this study, we cannot fully address concerns related to judgment biases in the grading process or standardized training of experts, which are common with synthetic methods, we have referred the readers to other studies that have treated these in more details: lines 336 – 337 of the revised manuscript reads "More details on the synthetic what-if analysis are given by Penning-Rowsell et al. (2005), Neubert et al. (2008) and Naumann et al. (2009)". Also, lines 578 – 587 further discuss on context-specific definitions of damage grades. These concerns will be made clearer in the revised manuscript.

Fourthly, the three phases in Section 5.2. could be better embedded in the overall picture of the paper. For example, Table 4 presents a damage grade scheme which is unclear whether the conceptual framework applies the same grading scheme or not. In Section 5.1 and Section 3.1, recommendations for best-practice are mentioned by Blong (2003b). I do not see these recommendations picked up in the framework.

Authors: Thank you for the comment. We have adapted sections 2 and 3 to reflect the methods used in section 5.2; we focused the sections on expert-based methods that can be applied in data-scarce regions. Sections 2 and 3 are streamlined and reorganized into a similar format for vulnerability indicators and damage grades by focusing on (1) background, (2) application, and (3) challenges for data-scarce regions. Further, an extended discussion on identified challenges are presented in section 4.

Table 4 serves to present an example of damage grades developed for Germany by Maiwald and Schwarz (2007). The conceptual framework recommends that damage grades are developed in a regional context. Frequently observed damage patterns within the region are to be used for developing the damage grades. In Section 5.2.2 we outlined that the main aim of this step is to identify commonly-observed damage patterns within a region and categorize them into classes.

The damage grades developed by Maiwald and Schwarz (2007), which our study was based upon, were developed using the recommendations by Grünthal (1993). Consequently, the techniques described in section 5.2.2 systematically integrates the recommendations. We will clarify this in a revised version of the manuscript.

Fifthly, the structure needs attention and the arguments are sometimes not placed in the right sections. In Section 2 you have too many (sub)subsection, followed by many lists with detailed arguments. Section 4 discusses the need for linking indicator and damage grades but is not clear whether it is linked to the own contributing or written as a conclusion of the literature review. In Section 5, the author's contribution should be in the center. Explaations and smaller reviews should be avoided here. I see Section 5.1. a bit like a repetition of what is explained in section 2, 3 and 4.

Authors: Thank you for the comment. Sub(sections) and details in section 2 have been reduced in the revised manuscript. Section 4 is not a direct conclusion of the reviews presented in sections 2 and 3. Rather, section 4, draws from challenges highlighted in the individual sections and illustrates the added value for combining them in data-scarce regions. This illustration was carried out by using three observed damage cases from a 2017 flood event in Nigeria. For section 5, the information provided has been reduced in the revised manuscript so that the content focuses on our contribution. Section 5.1 was not a repetition but rather background information for terminologies (e.g., vulnerability, exposure, susceptibility, local protection) we adopt in the concept. This information was necessary before introducing section 5.2 "Operationalizing the framework". The information was not provided in sections 2 and 3 (since they are reviews from different studies) or in section 4 (since it focuses on the linkage of vulnerability indicators and damage grades). We will make necessary adjustments on the revised version of the manuscript, see also comments to referee #1.

Sixthly, the main contribution of the conceptual framework, which I think is the applicability in data-scare regions is not sufficiently discussed in section 2 and 3. It should be more on the point. Also, the term 'adaptability' of physical vulnerability assessments to other regions could be better picked up in the review. Are all physical vulnerability assessments adaptive regional models? Which are regional adaptive? Why? I also suggest providing more information about the specific requirements and capacity for applying them across different regions. Your tables should reflect this by focusing on these aspects.

Authors: Thank you for your comment. We will streamline the contents of sections 2 and 3 to better address applications of each method for data-scarce regions. The suggested modification includes introducing challenges (for data-scarce regions) in both sections 2.3 and 3.3. These will serve as a background for research gaps that will be taken up and addressed in the framework. Another suggested change is to focus the review in section 2 on the deductive and normative methods since they are more suitable for data-scarce areas. Based on recommendations from referee #1, we will change the term 'adaptable' to 'generic' in the title. This was to avoid confusion with the adaption component of the Disaster Risk Reduction (DRR) literature. We will further address the need for making the indicators and damage grades context-based.

**"specific comments"**

Paper is too long and own contribution is relatively short.

Authors: Thank you for your comment. Substantial effort has been made to streamline and reduce the contents in sections 1-5 of the manuscript. Our contributions are also more highlighted in the revised manuscript.

Title: "adaptive" and "regional" are not well addressed in the paper.

Authors: Based on a recommendation from Referee #1, we have replaced the term 'adaptive' with the word 'generic' in order to avoid confusion with the adaptation component frequently used in the DRR literature. We will better address adaptability to the regional situation in the revised version of the manuscript.

Abstract: When reading the abstract the conceptual framework is in the center, however, this is not reflected by the paper which focuses more on the literature review.

Authors: Thank you for your comment. The abstract has now been reformulated to balance both the review and conceptual framework. The revised abstract now reads "The use of different methods for physical flood vulnerability assessment has evolved over time, from traditional single-parameter stage-damage curves to multi-parameter approaches such as multivariate or indicator-based models. However, despite the extensive implementation of these models in flood risk assessment globally, a considerable gap remains in their applicability to data-scarce regions. Considering that these regions are mostly areas with limited capacity to cope with disasters, there is an essential need for assessing the physical vulnerability of the built environment and contributing to an improvement of flood risk reduction. To close this gap we propose to link approaches with reduced data-requirements such as vulnerability indicators (integrating major damage drivers) and damage grades (integrating frequently observed damage patterns). First, we present a review of current studies on physical vulnerability indicators and flood damage models comprising stage-damage curves and the multivariate methods, which have been applied to predict damage grades. Second, we propose a new conceptual framework for assessing the physical vulnerability of buildings exposed to flood hazards specifically tailored to use in data-scarce regions. This framework is operationalized in three steps, (i) developing a vulnerability index, (ii) identifying regional damage grades, and (iii) linking resulting index classes with damage patterns utilizing a synthetic what-if analysis. The new framework is a first step for enhancing flood damage prediction to support risk reduction in data-scarce regions. It addresses selected gaps in literature by extending the application of the vulnerability index for damage grade prediction through the use of a synthetic multi-parameter approach. The framework can be adapted to different data-scarce regions and allows integrating possible modifications of damage drivers and damage grades".

Introduction: too long and broad

Authors: Thank you for your comment. The introduction has been streamlined and reduced in the revised version of the manuscript.

Line 81: Unclear how social loss is defined. I do not agree that building damages are the only or one of the most important factors for social loss. In particular, if you consider that not all affected people own a building.

Authors: Thank you for your comment. In the revised manuscript, we have excluded this part in order to streamline and reduce the manuscript. The introduction is now strongly focused on physical vulnerability.

Line 83: There is a critical difference between social and physical vulnerability assessment. You need to make clearer.

Authors: Thank you for your comment. In the revised manuscript, we have excluded this part in order to streamline and reduce the manuscript. The introduction is now strongly focused on physical vulnerability.

Line 143: more references.
Authors: Thank you for your comment. Changes have been made accordingly.

Line 149: the term 'holistic' needs a proper definition.
Authors: Thank you for the comment. The term 'comprehensive' will be used to denote an assessment that considers all possible influencing parameters.

Line 256: difference between indicator and index is not defined.
Authors: Thank you for your comment. Suggested modification in the revised manuscript, line 117 – 119, reads "A vulnerability index is obtained by selecting, weighting and aggregating vulnerability indicators. A vulnerability indicator is a parameter (or variable) that can influence and(or) communicate the vulnerability of a system (e.g., building). Generally, the aim of the indicator approach is to simplify a concept through the use of an index (Heink and Kowarik, 2010; Hinkel, 2011)".

Line 264: it is not objectivity what you mean it is comprehensiveness. Objectivity is needed for every selected indicator.
Authors: Thank you for your comment. Changes have been made accordingly.

Line 222ff. Indicator weighting: statistical weighting based on data can explain the consistency and inference of indicators but cannot be used for an appropriate weighting of importance or measurability of the indicators. This should be mentioned at the beginning.
Authors: Comment is not very clear. We will add an explanation of the indicator weighting in the revised version of the manuscript.

Line 571: unclear if this is part of the framework or part of literature review.
Authors: Section 4 focuses mainly on the added value for linking vulnerability indicators and damage grades for data-scarce regions. It is not meant to be part of the review or the new conceptual framework, rather, it is the authors' contribution to illustrate the added value of combining different physical vulnerability assessment methods. This was demonstrated by using a hypothetically developed vulnerability index for two regions in combination with three building damage data from Nigeria. This section uses practical examples to bridge the gap between identified challenges/gaps presented in the reviews (sections 2 and 3) and the new conceptual framework (section 5).

Line 701: Do you applied field surveys or remote sensing? When discussion about different option should be mentioned in the review section.
Authors: Carrying out a field survey to determine building typology and characteristics will be the most preferred option for such data collection because experts can have a first-hand impression of the local situation and can identify, in detail, construction features or qualities, which will determine how building representatives are selected. However, due to time and financial constraints, field surveys are not always feasible. Hence, we suggest the use of remote sensing as an option, especially in a meso-(or macro-) scale studies. The authors recommended the study by Blanco-Vogt and Schanze (2014) which focus on extracting building representatives for meso-(or macro-) scale assessment.

Conclusion: Be more precise about the key messages both from the review and from the conceptual framework. An outlook of how the framework will be applied is also helpful for presenting the relevance. Please elaborate on the link between the relevance for risk reduction methods in developing countries and the data scarcity and barriers in collecting data in most of the developing countries?
Authors: Thank you for your comment. In the revised manuscript, more details on challenges for data-scarce regions are now given in sections 2.3 and 3.3. The relevance of the linkage is also elaborated in section 4 of the revised manuscript (see also suggestions of referee #1). The conclusion is streamlined to be more precise. A short outlook for the model has been added in lines 648 – 650, which reads "Its applicability for predominant building types, such as the sandcrete block and clay buildings in Africa, has the potential to promote disaster mitigation in such regions. The application of the new framework to evaluate and compare model performance with a data-driven model is also encouraged. Such an analysis will communicate the success of the framework and also allow for further improvement."

**"technical corrections"**

Line 249: this sentence needs a reference
Authors: Thank you for your comment. Changes have been made accordingly.

Line 322: acronyms are not defined in table figure caption.
Authors: Thank you for your comment. Changes have been made accordingly

Line 329f. Sentence is unclear also the example does not seem to make sense.

Authors: Thank you for your comment. The example has been rephrased in the revised manuscript in order to better capture the intended purpose. The suggested change, now contained in line 211 – 217, is "For example, if we assume the same hazard level affecting both reinforced concrete and clay building, it is most likely that the clay building will incur higher damage (see Maiwald and Schwarz, 2012). Therefore, based on such data, experts may weight a reinforced concrete building as less vulnerable than the clay building. However, assigning a value that qualifies the extent to which the reinforced concrete building is less vulnerable than clay building (e.g., moderate, high, very high) requires expert knowledge. Such expert knowledge will likely come from information on the quality of regional construction types, material or local protection".

Line 443: reference missing

Authors: Thank you for your comment. Changes have been made accordingly.

Linen 415f:

Authors: The comment is not clear.

Line 650, 661 and 680: fourth level of headline should be avoided.

Authors: Thank you for your comment. Changes have been made accordingly.

[revised manuscript text omitted]

▪ Building material
▪ Building condition
▪ Distance to channel
▪ Flood depth | Sandcrete block
Moderate
100 m
1 m | Sandcrete block
Good
50 m
1.2 m | Clay
Poor
< 20 m
0.60 m |
| **REGION B: Aggregated Index** | 0.5 | 0.4 | 0.7 |
| Vulnerability indicators
▪ Building age
▪ Building quality
▪ Sheltering effect
▪ Flood depth | < 10 years
Good
Complete
1m | 20 years
Moderate
Partial sheltering
1m | > 30 years
Poor
No sheltering
1m |
| **Maiwald and Schwarz (2015)**
5-category damage grade
Germany | **DG 2**
Slight cracks in supporting element | **DG 4**
Partial collapse of supporting element | **DG 5**
Collapse |
| **Ettinger et al. (2016)**
4-category damage grade
Peru | **Light**
Signs of impact | **Heavy**
Partial/Total collapse | **Heavy**
Total Collapse |

[revised manuscript text omitted]

---

## Author Response (AR2)

**Authors**: We would like to thank the reviewer for his/her insightful comments. Below, answers to the concerns raised are provided point-by-point.

**Reviewer comment**: The paper has improved from the first version. However, it still needs to be better targeted to the problem of data scarce regions.

Section 2 still presents excessive details and information about indicators selection, development and weighting which should not be part of the paper as there is a vast amount of literature that is already discussing in details these issues. Contour information on the technicalities of designing indicators need not be part of the paper. It is also paradoxical to go into the details of indicators choice, design and development when the focus is on data scarce regions.

**Author response**: Thank you for your comment. We have further streamlined the content of section 2.2, in particular by deleting redundant material. However, given the reasons below, we strongly believe that the remaining text is important to understand our theme and to guide the reader through the overall manuscript content.

The details provided in section 2.2 are important for (i) unveiling the gaps presented as a result of our review, and (ii) presenting the application of the new concept of vulnerability assessment in data-scarce regions.

    i)       Reviews on vulnerability indicators in literature have so far been focused on a variety of topics, such as climate change-vulnerability and adaptive capacity (e.g., (Füssel, 2007; Hinkel, 2011), community disaster resilience (e.g., Asadzadeh et al., 2017)), dynamic flooding (e.g., Papathoma-Köhle et al., 2017), and multi-hazards (e.g., Kappes et al., 2012) such as a combination of flood and earthquake hazard assessment (e.g., De Ruiter et al., 2017). In our study, however, we provide one of the first of such reviews on indicators for physical vulnerability assessment in data-scarce regions, focusing on hydro-meteorological hazards.

    ii)      We have removed the information regarding inductive approaches because these approaches are dependent on data availability as stated by your comment. Thus, we take up the recommendation to focus on data-scarce regions with the essential details on deductive and normative methods which are applicable in these regions. These approaches allow also to apply the indicator based approaches in data-scarce regions and therefore considering the aim to provide a comprehensive review, it is in our opinion crucial to include the essential information on indicators for data-scarce regions.

    iii)     Furthermore, details provided in Section 2 on the development of vulnerability indicators is essential for implementing the proposed concept. The details on indicator selection (preliminary and final), weighting and aggregation provide the basis for Phase 1 of the conceptual framework. The Phase 1 in turn provides the basis for the systematic vulnerability classification (BRI classes) proposed and serves as a fundamental step for the application of Phases 2 (development of damage grades) and 3 (what-if analysis).

**Reviewer comment**: The innovative aspects and the features of the approach specifically tailored to data scarce regions are difficult to judge.

**Author response**: Thank you for your comment, which shows us that the central theme again needs some clarification. We have added lines 628 – 631 to provide additional information on the innovative aspects of the proposed method.

In addition, in Section 4 the focus specifically is on benefits of the method for physical vulnerability assessment in data-scare regions. The value of these contributions is highlighted in lines 416 – 428 and quoted below.

"A combination of physical vulnerability indicators and damage grades using the synthetic approach has a number of advantages for data-scarce areas. These include:

    I.     Employing the synthetic what-if analysis to link damage grades and damage drivers allow to overcome high data requirements of the multivariate method. Consequently, the linkage will capture multiple damage-influencing variables. Also, using the damage grades will allow to carry out performance checks on the effectiveness and robustness of selected vulnerability indicators.

    II.    The linkage will enable to compare consequences of flood hazards across spatial and temporal scales in data-scarce regions. Spatial comparability can be achieved through the identification of similar damage characteristics (Fig. 2) between regions with similar building types and hazard characteristics. Temporal comparability can be achieved by relating the severity of observed damage grades between different flood events since damage grades are not readily affected by market values or wages. In addition, using similar hazard scenarios damage can be estimated and compared between regions while still considering individual damage drivers (Fig. 2).

    III.   Since damage grades are physically observable features, the linkage will foster the provision of an easy communication tool for stakeholders and community residents on the consequences of hazards"

**Reviewer comment**: Finally, in the first section you mention that you present discussion and conclusions in section 6 but there is only a conclusions section 6 which is long and appears as a summary of the approach rather than a proper concluding section. Instead best practices could be given on what we learned regarding how to handle the measurement of physical vulnerability in data scarce regions.

**Author response**: Thank you for your comment. We changed the conclusion section in order to better mirror the manuscript content and to provide a more appropriate outlook for further studies. The section is now titled "Concluding remarks" with the following content:

[revised manuscript text omitted]

▪ Building material
▪ Building condition
▪ Distance to channel
▪ Flood depth | Sandcrete block
Moderate
100 m
1 m | Sandcrete block
Good
50 m
1.2 m | Clay
Poor
< 20 m
0.60 m |
| **REGION B: Aggregated Index** | 0.5 | 0.4 | 0.7 |
| Vulnerability indicators
▪ Building age
▪ Building quality
▪ Sheltering effect
▪ Flood depth | < 10 years
Good
Complete
1m | 20 years
Moderate
Partial sheltering
1m | > 30 years
Poor
No sheltering
1m |
| **Maiwald and Schwarz (2015)**
5-category damage grade
Germany | **DG 2**
Slight cracks in supporting element | **DG 4**
Partial collapse of supporting element | **DG 5**
Collapse |
| **Ettinger et al. (2016)**
4-category damage grade
Peru | **Light**
Signs of impact | **Heavy**
Partial/Total collapse | **Heavy**
Total Collapse |

[revised manuscript text omitted]